# KUIPER: MODERATED ASYNCHRONOUS FEDERATED LEARNING ON HETEROGENEOUS MOBILE DEVICES WITH NON-IID DATA

## ABSTRACT

Federated learning allows multiple clients to jointly learn an ML model while keeping their data private. While synchronous federated learning (Sync-FL) requires the devices to share local gradients synchronously, to provide better guarantees, it suffers from the problem of stragglers, slowing the entire training process. Conventional techniques completely drop the updates from the stragglers and lose the opportunity to learn from the data the stragglers hold, especially relevant in a non-iid setting. Asynchronous learning (Async-FL) provides a potential solution to allow the clients to function at their own pace, which typically achieves faster convergence. We target the video action recognition problem on edge devices as an exemplar heavyweight task to perform on a realistic edge setup using asynchronous-FL (Async-FL). Our FL system, KUIPER, leverages Async-FL to learn a heavy model on video-action-recognition tasks on a heterogeneous edge testbed with non-IID data. KUIPER introduces a novel aggregation scheme, which solves the straggler problem, while taking into account the different client data in a non-iid setting. Although the proposed aggregation technique is catered majorly for video action recognition, it is task-independent and scalable, and we demonstrate it by showing experiments on other vision and NLP tasks. KUIPER shows a 11% faster convergence compared to Oort [OSDI-21], up to 12% and 9% improvement in test accuracy compared to FedBuff [AISTAT-22] and Oort [OSDI-21] on HMDB51, and 10% and 9% on UCF101.

## 1 INTRODUCTION

Federated learning McMahan et al. (2017) has gained great popularity in recent times as it allows heterogeneous clients to collaborate and benefit from peer data while keeping their own data private. As a result, the clients learn a better model with collaboration than they would have, individually. The training process is orchestrated by a central server that broadcasts the global model to the clients while the clients run local training on their own data and only share the gradient updates with the server. This has made it possible for clients with limited computational resources to participate in the learning process. However, heterogeneous clients with varying computational capabilities (we use the term "computational capabilities" as a shorthand to include heerogeneity in both computational capabilities on the node as well as the communication capabilities connecting the node to the federation server), if forced to synchronize, direct the process to progress at the speed of the slowest client Li et al. (2020a). For example, in our experimental setup of embedded nodes with mobile GPUs, Jetson Nano is $5\times$ slower than Jetson AGX Xavier; including variation in network speeds adds to this heterogeneity. It becomes crucial to incorporate even slow clients when the data distribution among clients is non-IID, as all clients then have distinctive elements to contribute to the learned model.

In this paper, we target a heavyweight learning task, namely, video action recognition, that till date had been considered out of the reach of embedded devices, *i.e.*, mobile GPUs. The straggler problem becomes particularly serious for heavyweight learning tasks on heterogeneous edge devices since the devices are resource constrained relative to the demands of the task and the variance in device capabilities (processing power, memory, storage) is large ($5\times$ in our representative setup). Therefore, to deal with stragglers an obvious approach seems to be to use synchronous learning. However, this prevents the global model from learning features specific to the local data of the stragglers, leading to a model that underfits. This problem becomes more acute as the degree of non-IIDness increases;

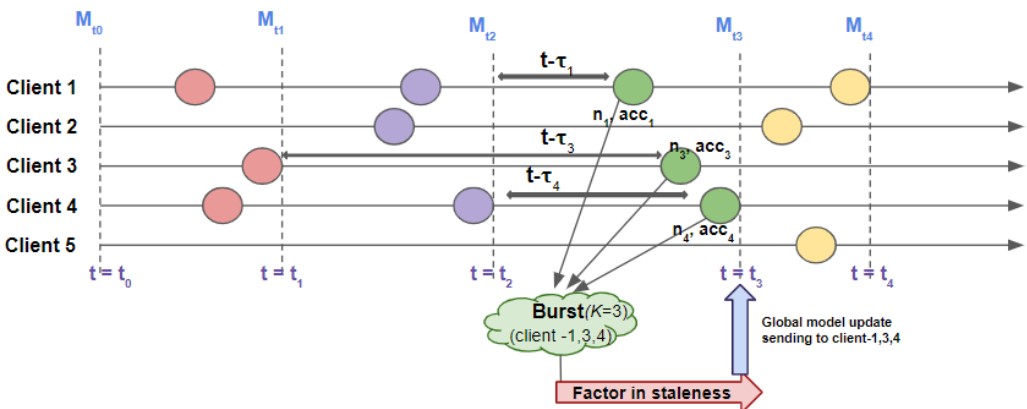

**Figure 1:** Overview of a working example of KUIPER in action for 5 heterogeneous clients with $K$ (burst size) =3. The circle denotes that a client is ready with its updates. The dashed vertical line denotes an aggregation step where we also update $\tau_i$ for the clients aggregated in the burst. The aggregator waits for 3 clients to respond, comprising a burst, denoted by identically colored circles. Within the burst, the individual client updates are weighed by a function of their local data size and training accuracy. The burst, as a whole, is then weighed again by the average staleness $(t - \tau_i)$ of the clients comprising the burst, and the global model is updated. The updated model is sent back to all the clients in that same burst, and the process goes on.

again, for a distributed edge device scenario, high degrees of non-IIDness are commonly seen Zhao et al. (2018); Chen et al. (2020b). We empirically observe the severe negative consequence of discarding stragglers on the learning accuracy (Figure 7(d)). This motivates the use of asynchronous aggregation, which allows the central server to aggregate the clients' gradient updates as soon as they are made available without having to wait for all the clients to respond. *However, it has remained an open problem how to best aggregate the updates sent by all clients in order to maximize information learned while minimizing any adverse effect from slow updates*[1].

**Our proposed solution KUIPER:** We propose KUIPER[2] to solve the above problems of heterogeneous clients with resource constraints and non-IID data, with the overview shown in Figure 1. We consider the typical case of FL with non-IID data where although the client might not have training data for all the classes but wants to have a global model which can work on all the classes (i.e., learning from peers). Our solution is based on the idea of scaling the stale updates before aggregation, depending on the staleness of the updates, and the current iteration's training error of the clients. Training error is a measure of how much the local model has made progress on learning from its own data. This ensures that the global model is not starved of the information that could be learned from the stragglers' data. Our scaling policy is designed to ensure high model quality while balancing the need to incorporate relatively outdated updates if they improve the global model. Further, we find that a pure asynchronous solution does not work well due to the wide diversity of rates of client updates. We then batch the updates from a group of clients, quantified by $K$, the *batch size*, before aggregation. This makes KUIPER a buffered asynchronous approach[3].

Our contributions can be summarized as follows:

1. We propose a novel scheme to include heterogeneous clients in federated learning by balancing the utility of their data with their computational (and communication) efficiency.

2. We demonstrate our heterogeneous FL technique through video action recognition, which is a computationally heavy task and can be accomplished on resource-constrained edge devices only through the use of FL. In our setting, this task is particularly challenging due to device heterogeneity, network heterogeneity, and non-IID data.

---

[1]There are two recent promising solutions to this problem in Oort Lai et al. (2021) and FedBuff Nguyen et al. (2022), and we discuss why they fall short and also compare them empirically to our solution.

[2]KUIPER is a band of small celestial bodies beyond the orbit of Neptune from which many short-period comets are believed to originate. Similarly, we make the small devices coalesce to achieve big tasks.

[3]Aspects of this design are shared with FedBuff Nguyen et al. (2022); we explain the differences in Section 2 and empirically demonstrate our superiority (Section 5).

3. We provide a convergence analysis and show the effect of the number of clients in the federation and the non-IID parameter on the convergence.

4. With a comprehensive evaluation of our proposed design on three Video Action Recognition datasets (Kinetics, HMDB51, and UCF101), we also show scalability of our algorithm on two other tasks (Image Recognition, and Next-Character Prediction). We provide insights on the relationship between data staleness and non-IIDness and show how KUIPER achieves, for the HMDB51 dataset, up to a 12% improvement in the action recognition task's test accuracy compared to FedBuff Nguyen et al. (2022), and 9% compared to Oort Lai et al. (2021), and an improvement of 10% and 9% respectively for the UCF101 dataset. Note that action recognition is a challenging task and even centralized training *does not* reach accuracy of 50% (47.8% to be exact) for a frame rate of 8, making the above (absolute) gains in accuracy significant.

## 2 RELATED WORK

FL McMahan et al. (2017) is used in multiple applications that require data privacy, for example, in healthcare Chen et al. (2020a); Li et al. (2019) and natural language processing on smartphones Llisterri Giménez et al. (2022); Leroy et al. (2019). In synchronous FL, the server waits to receive updates from all the clients before aggregating the global model. FedAvg McMahan et al. (2017) is the baseline synchronous FL, where the global model is updated by the weighted sum of the gradients, weighted by the size of the local client data. FedProx Li et al. (2020b), FedAvgM Hsu et al. (2019), FedAdam Mills et al. (2020), are all modifications made to FedAvg to speed up convergence, albeit all of these are synchronous FL techniques.

Asynchronous optimization, solves the problem of straggler clients. Its innovation lies in rewarding and penalizing the clients in terms of the usefulness or staleness of the updates as in Smith et al. (2017); Xu et al. (2019). FedAsync Xie et al. (2019) sets the local learning rate of clients to be inversely proportional to the frequency with which the client generates the model updates, in order to increase the contribution of slower clients. Oort Lai et al. (2021), on the other hand waits for a fixed number of clients ($K$) to synchronize and runs FedAVG McMahan et al. (2017) on the received updates. It prevents starvation of stragglers by maintaining a dynamic utility score for every participating client. This depends on the staleness and local training loss of the individual clients. FedBuff Nguyen et al. (2022) also waits for a fixed number of clients ($K$) to update its gradients but does not have a client selection policy or a gradient weighing policy according to their performance. Thus, the model is biased towards the fast clients' data distribution (as it gives more weight to the fast clients because of their more frequent updates). When the non-IID bias is high, some slow clients will have exclusive data, which is important for the overall training and thus need to aggregate that client's model with high importance. Its focus is on guarding against an honest-but-curious server (which it achieves by storing the buffer in a TEE) and in ensuring scalability to hundreds of clients (which is helped by the buffering).

Asynchronous FL, however has its own challenges. Previous literature in Asynchronous Learning Xie et al. (2019); Chen et al. (2020b) penalizes clients for their delayed updates and thus contribution of a slow client to the global model is curtailed. In such scenarios, the problem arises when the data that clients have is distributed in a non-IID manner, exacerbated for high non-IID bias values. In such cases, a few clients may possess useful data while being stragglers. Previous literature does not consider this aspect and thus performs poorly when the non-IID bias is high (as we empirically show with FedBuff in Figures 3 and 6). We have evaluated Xie et al. (2019)'s method and it has comparable, albeit lower, accuracy relative to KUIPER for IID data. However, KUIPER's performance is higher than its counterparts for highly skewed data distributions, as would be the case in realistic mobile computing devices.

The fact that edge devices are often constrained in terms of local resources (compute, memory, and storage) as well as network resources (low bandwidth connections, intermittent connectivity) has given rise to fruitful areas of inquiry in communication-efficient federated learning Reisizadeh et al. (2020); Sattler et al. (2019); Mills et al. (2019), asynchronous learning to deal with stragglers Smith et al. (2017); Xu et al. (2019), approximate models and computation Zhang et al. (2018); Wu et al. (2019); Han et al. (2020), and also knowledge distillation to create more succinct models Jang et al. (2020); Matsubara et al. (2020). Federated distillation Jeong et al. (2018) follows an online version of knowledge distillation, known as co-distillation (CD) Anil et al. (2018). In CD, each device treats

itself as a student, and sees the mean model output of all the other devices as its teacher's output. Furthermore, non-IID data of on-device ML can be corrected by obtaining the missing local data samples at each device from the other devices. This can induce significant overhead, so FAug Jeong et al. (2018) is proposed. FAug generates the missing data on each device. They empirically found that their approach yields lower overhead and better accuracy for image classification on MNIST LeCun et al. (1998). Human action recognition approaches can be categorized into visual sensor-based, non-visual sensor-based, and multi-modal categories Yurur et al. (2014); Ranasinghe et al. (2016). So far, federated learning for action recognition has only been incorporated into federated learning using wearable sensors Sozinov et al. (2018); Ek et al. (2020). That is an easier task since the data streams from these sensors are much lighter compared to the targeted video data.

## 3 DESIGN AND ANALYSIS OF KUIPER

KUIPER is a buffered asynchronous aggregation technique, which is designed considering the non-IID biases in the clients' datasets and heterogeneity in clients' computational resources.

### 3.1 DESIGN DETAILS

The global server node and the client nodes conduct the training in a buffered asynchronous manner. Each client independently trains the model obtained from the server on its local data and shares the gradient update with the server as soon as it is ready. The server does not wait to hear from *all* the clients. Rather, the aggregation works in bursts where a burst consists of $K$ clients that have responded and are waiting for the server to send them back the aggregated global model. The server aggregates the received updates according to the client's local data size and training accuracy, and their staleness. It then updates the global model with these aggregated gradients and sends it to only those clients that contributed to the burst. Meanwhile, other clients might have responded to the server with their gradients and the server again waits until it has heard from $K$ clients to form a burst, and continue the above described process iteratively until convergence. We demonstrate a working example in Figure 1 with 5 heterogeneous clients and $K = 3$.

**Problem formulation.** We consider a federated learning setup with $M$ devices. We consider a supervised problem where the data is partitioned across $M$ different clients with $D_1, D_2, .., D_M$ data. Data samples are different for all the clients, *i.e.*, $D_i \cap D_j = \phi$ for all the i, j $\in [M]$ and $\bigcup_{i=1}^{M} D_i = D$, where $D$ is the complete training data. Our aim is to find the parameters $w$ that achieves $\min F(w)$, i.e.

$$w_{opt} = \min_w F(w), \text{where} F(w) = \frac{1}{M} \sum_{k=1}^{M} \mathbb{E}[l(w; d_i)] \tag{1}$$

Here, $d_i$ is data sampled from local data $D_i$ on the $i$-th device, and $l(\cdot; \cdot)$ is a user-specified loss function. The $i^{th}$ client performs training with a learning rate $\eta_l$ using data $d_i$, which is randomly sampled from its local dataset $D_i$. We consider the typical case of FL with non-IID data where although the client might not have training data for all the classes but it wants to have a global model which can work on all the classes (i.e., learning from peers).

**Knowledge distillation.** To accommodate the limited resources on the embedded devices, we use knowledge distillation to train a light-weight model ResNet-18, initialized from ResNet-34, trained on the Kinetics dataset. We define the knowledge distillation loss $L_{KD}$ as the Mean Squared Error between the logits from the teacher model $\boldsymbol{z}^t$ and the student model $\boldsymbol{z}^s$, *i.e.*, $L_{KD} = \|\boldsymbol{z}^t(x) - \boldsymbol{z}^s(x)\|^2$. The overall loss function is a combination of two loss functions, $L = \alpha L_{cls} + (1 - \alpha)L_{KD}$. $L_{cls}$ is the conventional cross-entropy loss, computed for the predictions made by the student and the ground truth corresponding to the input $\mathbf{x}$.

The teacher model cannot effectively transfer its knowledge to the student if the size gap between them is large Mirzadeh et al. (2020). To alleviate this, the knowledge distillation is done through an intermediate Teaching Assistant (TA) model, which in our case is ResNet-26.

**Fine tuning at the clients.** In every epoch, the central server waits for $K$ clients to report their updates, with these $K$ clients forming a burst. The individual gradients from each client within the burst are weighed according to three factors and shown in Equation 2: the amount of data at each client, the current training accuracy at the client, and the speed of the client. For larger non-IID bias, clients have data only from a subset of classes, and thus their reported gradients become relatively noisy. Weighted-averaging those gradients first in a burst and then aggregating the burst with the

global model helps to achieve a better accuracy. Averaging also helps prevent inference attacks, as mentioned in Nguyen et al. (2022). Later, we experimentally see the importance of this buffering over the vanilla asynchronous mode for video action recognition (Figure 9 (a)).

Now let us look at the various components of Equation 2.

$$w_{new,t}^{c_t} \leftarrow \sum_{i=1}^{K} \frac{n_i}{N} w_{new,t}^{i} \{ \mathbf{1}(t < T_0) \times e(acctrain_t^i) + \mathbf{1}(t \geq T_0) \} \tag{2}$$

The term $n_i/N$ normalizes each client by the amount of data that it has. $\mathbf{1}(\cdot)$ is the identity function, which is 1 when the argument is True else 0. This rewards clients which return results within a latency threshold $T_0$. The function $e(acctrain_t^i) = 1 - acctrain_t^i$, considers this and thus give more importance to the clients that have a low training accuracy with the current state of the model. Intuitively, when a client's training accuracy is high, it means that the global model has already learned the features corresponding to that client's data and our global model can focus on other clients to learn their features.

Now let us consider how the aggregation handles stragglers and penalizes stale updates from clients. This is achieved in the second level of aggregation, where the weighting factor of the burst is determined (Equation 3).

$$\beta_t^{c_t} \leftarrow \beta \times s(t - \tau^{c_t}) \tag{3}$$

Here, $c_t$ is the set of clients ($\{1, 2, .., i, .., K\}$) we are considering in the $t^{th}$ update of the model. We calculate $w_t^g$ the global model at epoch $t$ using Equation 4.

$$w_t^g \leftarrow (1 - \beta_t^{c_t})w_{t-1}^g + \beta_t^{c_t} w_{new,t}^{c_t} \tag{4}$$

To do this, we moderate the mixing hyperparameter, $\beta \in (0, 1)$. Here $t - \tau^{c_t}$ captures how delayed the burst is and we calculate staleness of the burst as $s(t - \tau^{c_t}) = (1 + t - \tau^{c_t})^{-\alpha}$, where $\tau^{c_t} = avg\{\tau^i, \forall i \in (1, 2, .., i, .., K)\}$, which adaptively changes the mixing parameter $\beta_t^{c_t}$. The general form of this function is that it monotonically and exponentially decreases with increase in staleness. The above is presented as a pseudo-code in Algorithm 1 in Appendix B.

## 3.2 CONVERGENCE ANALYSIS

Here we prove the convergence guarantee of KUIPER. This analysis is influenced from FedBuff Nguyen et al. (2022) and customized to our model. Specifically, we characterize the effect of non-IID bias on gradient variances and convergence guarantee.

**Notation.** M denotes total number of clients. $g_i(w; \zeta_i)$ denotes stochastic gradient on $i^{th}$ client on a model with weights $w$ and sampled batch $\zeta_i$. $\nabla F_i(w)$ denotes the gradient with respect to the loss. $\sigma_l^2$ and $\sigma_g^2$ are local and global variances of the gradients. $f(w)$ is the objective function and $f^*$ is the theoretical minima. $t$ is the current iteration and $\tau_i$ is the global iteration when $i^{th}$ client received gradients from the server.
**Assumption 1:** (Unbiased client stochastic gradients) $\mathbb{E}[g_i(w; \zeta_i)] = \nabla F_i(w)$.
**Assumption 2:** (Bounded local and global variance) $\forall i \in [M], \mathbb{E}_{\zeta_i|i}[||g_i(w; \zeta_i) - \nabla F_i(w)||^2] \leq \sigma_l^2$ and $\frac{1}{M} \sum_{i=1}^{m} ||\nabla F_i(w) - \nabla f(w)||^2 \leq \sigma_g^2$.
**Assumption 3:** (Gradients are bounded) $||\nabla F_i||^2 \leq G$.
**Assumption 4:** (L-smoothness), $\forall i \in [M]$, the gradient is L-smooth, $||\nabla F_i(w) - \nabla F_i(w')||^2 \leq L||w - w'||^2$.
**Assumption 5:** (Bounded Staleness) The staleness of stragglers $t - \tau$, where t represents current global epoch and $\tau$ represents the global epoch when the client last synchronized with the server, is bounded $t - \tau \leq \tau_{max,1}$ which is the maximum across all the clients.

Choosing a constant local learning rate $\eta_l$ and global learning rate $\eta_g$ such that $\eta_g \eta_l Q \leq \frac{1}{L}$, the global model iterates in KUIPER are bounded by

$$\frac{1}{T} \sum_{t=0}^{T-1} \mathbb{E}[||\nabla f(w^t)||^2] \leq \frac{2F^*}{\eta_g \eta_l Q T} + \frac{L}{2} \eta_g \eta_l \sigma_l'^2 + 3L^2 Q^2 \eta_l^2 (\eta_g^2 \tau_{max,K}^2 + 1)\sigma'^2 \tag{5}$$

where $F^* := f(w^0) - f^*$, $\sigma'^2 := \sigma_l'^2 + \sigma_g'^2 + G'$. $\sigma_l'$ and $\sigma_g'$ are the new bounds of local and global variance, and $G'$ the updated norm of gradients when the gradient updates are scaled by $s(\cdot)$ and $e(\cdot)$. $Q$ is the number of local iterations for a client, and $T$ the total number of global iterations. Further, choosing $\eta_l = \mathcal{O}(1/(K\sqrt{TQ}))$ and $\eta_g = \mathcal{O}(K)$, for all $\eta_g, \eta_l$ satisfying $\eta_g \eta_l Q \leq \frac{1}{L}$ and sufficiently

large $T$, we have

$$\frac{1}{T}\sum_{t=0}^{T-1}||\nabla f(w^t)||^2 \le \mathcal{O}(\frac{F^*}{\sqrt{TQ}}) + \mathcal{O}(\frac{\sigma_l'^2}{\sqrt{TQ}}) + \mathcal{O}(\frac{Q\sigma'^2}{TK^2}) + \mathcal{O}(\frac{Q\sigma'^2\tau_{max,1}^2}{TK^2}) \qquad (6)$$

For sufficiently large $T$, the algorithm achieves the convergence rate as shown in Eq. (10). We provide a detailed proof in the Appendix I. As we can see, the convergence guarantee increases with increasing $K$ as we tend to go closer to the synchronous aggregation. Also, as non-IID bias increases, gradient variances increase, and weakens the convergence guarantee.

## 4 IMPLEMENTATION

The central server in the following experiments has an NVIDIA Tesla V100S 32GB GPU. We use four types of mobile GPU-equipped clients to demonstrate that our asynchronous federated optimization is robust to heterogeneous edge devices: *NVIDIA Jetson Nano*, which has 4GB memory, and a 128-core Maxwell GPU; *NVIDIA Jetson TX2*, which has 8GB memory, and a 256-core Pascal GPU; *NVIDIA Jetson Xavier NX*, which has a 8GB memory, and a 384-core Volta GPU with 48 Tensor cores; and *NVIDIA Jetson AGX Xavier*, which has 32GB memory and 512-core Volta GPU.

We use two different setups where $M$ (Number of clients) = 4 and $K$ (Burst size) = 2, and $M$=12 and $K$=4. For $M$=4, we use one device each from the above categories of devices, and for the $M$=12 setup, we use three devices each from the above categories.

The Kinetics Kay et al. (2017) dataset, which we use for knowledge distillation, is present at the central server. We conduct experiments on two datasets for finetuning: HMDB51 Kuehne et al. (2011) and UCF101 Soomro et al. (2012). This data is distributed amongst the clients. The Kinetics dataset contains 400 human action classes, with at least 400 video clips for each action. Each clip lasts for around 10s and is taken from a different YouTube video. The dataset has 306,245 videos, and is divided into three splits: one for training, with 250–1000 videos per class; one for validation, with 50 videos per class; and one for testing, with 100 videos per class. The HMDB51 dataset contains 51 classes and a total of 3,312 videos. The UCF101 dataset consists of 101 classes and over 13K clips (27 hours of video data). We use the HMDB51 dataset for all experimental purposes unless otherwise stated. The model was trained using a learning rate of 0.001, staleness penalty $\alpha$ of 0.5, mixing parameter $\beta$ of 0.7 (Appendix D), batch size of 8 video clips, for 200 global iterations with 3 local epochs per client. We show our algorithm's scalability to a large number of clients for image recognition and next-word prediction tasks with other methods.

## 5 EXPERIMENTAL EVALUATION

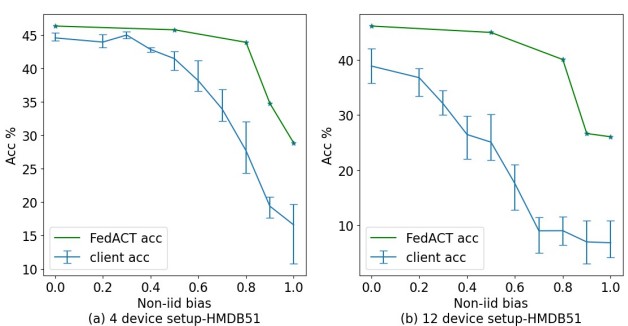

(a) 4 device setup-HMDB51      (b) 12 device setup-HMDB51

**Figure 2: Validation accuracy achieved by KUIPER as compared to individually trained clients across varying degrees of non-IID bias on HMDB51. Fig. (a) and (b) show the comparison on a 4-device and 12-device setup respectively, as described in Section 4. We observe that the improvement in accuracy achieved with KUIPER increases with higher degree of non-IIDness.**

In our evaluation, we ask, and answer, the following questions in order: (1) Is FL feasible for the heavyweight task of video action recognition on embedded devices? Is Knowledge Distillation useful for this? (2) How does KUIPER compare in terms of accuracy and time to train vis-à-vis the state-of-the-art in FL with heterogeneous clients, namely, FedAsync Xie et al. (2019), FedBuff Nguyen et al. (2022), and Oort Lai et al. (2021). (3) What is the effect of the Burst Size ($K$) on accuracy and time to train KUIPER and the two baselines, FedBuff and Oort? (4) What is the effect of a slow client, on KUIPER as well as the two baselines, FedBuff and Oort? (5) Ablation study of KUIPER showing the effect

of each of its components and the hyperparameters $\alpha$ and $\beta$.

**Is FL useful for action recognition, a computationally heavy task?** In this experiment, we motivate the use of FL in the action recognition scenario (the HMDB51 dataset). We first train each

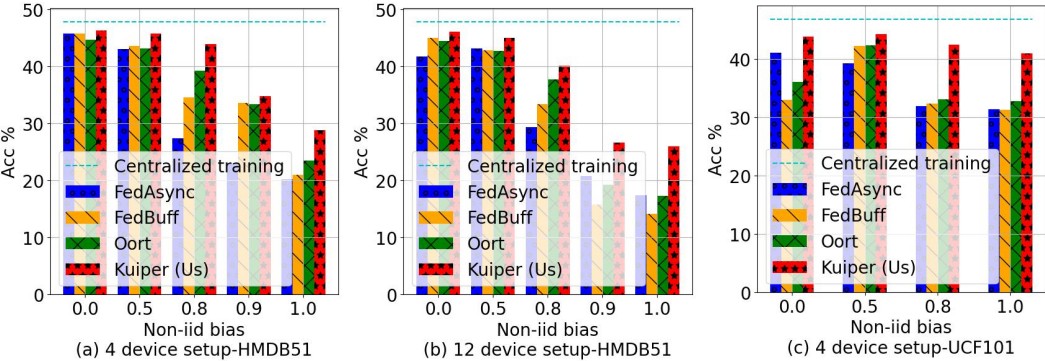

**Figure 3: Comparison of KUIPER with the FedAsync, FedBuff, and Oort baselines for varying non-IID bias. Figure (a) shows the comparison when the training was done on a 4-device setup with HMDB51 dataset. Figure (b) on a 12-device setup with HMDB51 dataset(see Section 4), and Figure (c) on a 4-device setup with the UCF101 dataset. KUIPER achieves a higher validation accuracy across all non-IID bias values in all the setups. The change in relative accuracy for KUIPER as compared to the baselines gets better with higher non-IID bias as we also carefully consider data quality. KUIPER outperforms Oort and FedBuff in absolute accuracy when non-IID bias=1.0 by (a) 5% and 8% (b) 9% and 12% (c) 9% and 10%.**

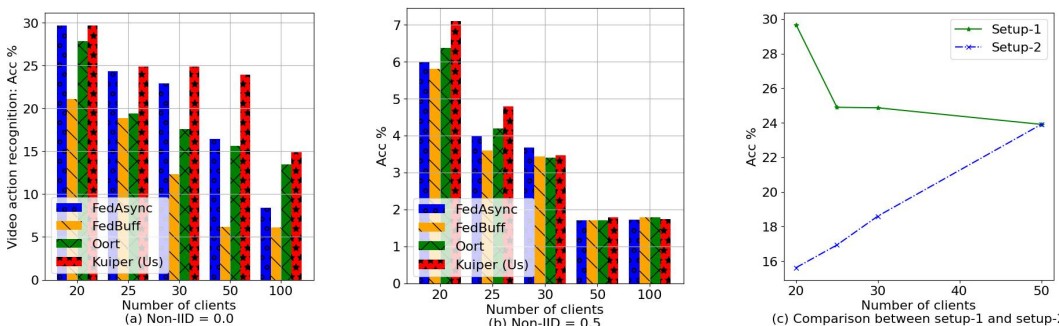

**Figure 4: Accuracy achieved with different number of clients on HMDB51 dataset . Here $K = 10$ is fixed across KUIPER, FedBuff and Oort. (a) non-iid = 0.0 and (b) non-iid = 0.5. KUIPER achieves higher accuracy than other baselines. For a non-iid value of 0.5 and with an increasing number of devices, achieved accuracy approaches to a random guess accuracy which is $\frac{1}{number\ of\ classes}$.**
**Accuracy achieved with different number of clients on HMDB51 dataset . Here $K = 10$ is fixed across KUIPER, FedBuff and Oort. (c) Comparison between Setup-1 and Setup-2. In Setup-1, increasing client means fewer samples per client; thus, accuracy decreases with the increasing number of clients. In Setup-2, increasing client means an increase in the training data, and thus accuracy increases with the increasing number of clients.**

client's model on its own data without collaboration for 50 epochs where the non-IIDness of the data distribution among the clients is varied. We report each client's validation accuracy in Figure 2 and compare it with that achieved with our aggregation technique in a buffered asynchronous FL setting. Error bars correspond to minimum and maximum individual accuracy among the clients and the curve shows the mean accuracy across the clients. We observe a clear improvement in accuracy when KUIPER is used, as compared to the accuracy of each client, motivating the use of FL. The improvement becomes more marked with higher non-IID bias. An improvement of up to 15% and 8% was observed for two setups involving 4 and 12 clients, respectively.

**Baseline comparison** Previous asynchronous aggregation methods like FedAsync Xie et al. (2019), penalize all the lagging clients uniformly without considering the data quality a client holds. This usually leads to under-utilization of a client's updates and the system suffers an accuracy drop. FedBuff Nguyen et al. (2022) does not consider the quality of the data that clients have. Oort Lai et al. (2021) considers both forms of utility of a client — how resource rich it is and how valuable is its data — to decide on client selection. However, once chosen, it gives the same weight to all clients' updates. With KUIPER, we appropriately balance the delay penalty and data quality reward and thus perform better than the three baselines for both HMDB51 and UCF101. Experiments are performed with 4 and with 12 devices, following the setup described in 4. We vary the non-IID bias and observe that the improvement over all baselines increases with increasing non-IID bias as shown in Figure 3.

**Figure 5: Scalability: Comparison of KUIPER with other methods on different tasks. Image recognition on a) MNIST b) FMNIST c) CIFAR10, and d) Next character prediction task on Shakespeare dataset.**

**Scalability: Large number of clients** In this section, to analyze the scalability of KUIPER, we propose two different setups: **Setup-1: Data samples per client decreasing with increasing number of clients**: Here, the total number of data samples is constant in every run, and with an increasing number of clients, we partition the dataset equally among the clients . For example, HMDB51 datasets have 8062 samples when we have eight frames per clip. When we consider four clients, each client gets 2013 data samples. For 100 clients, each client gets around 80 samples. With 80 data samples per client, we get around 1-2 samples per class when we have iid distribution. It gets challenging to train models on such small data samples, and we can see the achieved test accuracy decreasing with an increasing number of clients (Figure 4 setup-1). Due to this reason, we limit our analysis to a maximum of 100 clients. Figure 4 (b) validates how even with 50 clients and a 0.5 non-iid bias value, it is not possible to train the model as achieved accuracy is equivalent to random prediction accuracy.

**Setup-2: Data samples per client increasing with increasing number of clients**: In a real-world scenario, more clients bring more data and thus help learn a model with good feature representation, which we have tried to mimic in this setup. In this section, unlike decreasing samples, we first create 50 data slices from the total data. We assign one to every participating client. So, in this setup, if 20 clients participate, we have 40% of the total dataset in that training experiment. So, total training data increases with increasing clients as it should be in a real-world scenario. (Figure 4 Setup-2).

**Comparative performance on other tasks** In Figure 5, we show how KUIPER performs compared to other methods with up to **1,000** devices. Shakespeare dataset is used for the next-character prediction task. We have used perplexity loss (lower the better) for comparison (Figure 5 (d)). We used MNIST, FMNIST, and CIFAR10 datasets (Figure 5 (a, b, c)) for the image recognition task. We use accuracy as a metric for the comparison here. We thus see that KUIPER is a scalable solution and the advantage of KUIPER over the state-of-the-art baselines is maintained even at large scales.

**Effect of burst size ($K$) parameter** We show the effect of varying Burst Size ($K$) on KUIPER as well as the two baselines, FedBuff and Oort (Figure 6) (this is a 12-device experiment on HMDB51; UCF101 analyzed in the Supplement). As $K$ becomes higher, the protocols become closer to synchronous aggregation. We see that KUIPER outperforms others for any given value of $K$. Another way of looking at this is that to reach the same accuracy as KUIPER, FedBuff and Oort will need higher values of $K$. Consequently in Figure 6(b), we see that the time taken to reach a given accuracy is lowest in KUIPER. In Figure 6(c) we see that a synchronous approach like Oort takes much longer per aggregation round compared to FedBuff and KUIPER (KUIPER being slightly lower than FedBuff). This is due to Oort always waiting for the $K$ chosen clients in each epoch.

**Effect of stragglers** A straggler is a slow client and we incorporate their inputs in our aggregation technique by weighing the updates in accordance with their staleness and quality as described in Section 3. Figure 12 (a, b) compares the two setups where all four devices are homogeneous (NX)) vs. three devices are the same (NX), and one slow device (Nano) is there. Updates of this device are delayed and thus stale. With a slow device, accuracy decreases (Figure 12 (a)), and the time taken to reach a specific accuracy increases. Here, note that Oort waits explicitly for the $K$ clients depending on their utility scores, and waiting for the delayed client makes Oort slower than FedBuff even though it was faster in the homogeneous case (Appendix G).

Figure 7 (a, b) shows the analysis with 12 clients, 4 with no delay, 4 with $3\times$ delay, and 4 with $5\times$ delay — these delays are a multiple of the natural delay. The delay ratios have been chosen in order to mimic a realistic scenario. For example, the Jetson Nano device is $\sim 5X$ slower, and the Jetson TX2 device is $\sim 3X$ slower than Jetson AGX Xavier. For (a) the aggregation technique used is KUIPER in all the cases. Here, 8 with delay means, 8 devices are aggregated and 4 slowest are dropped. From

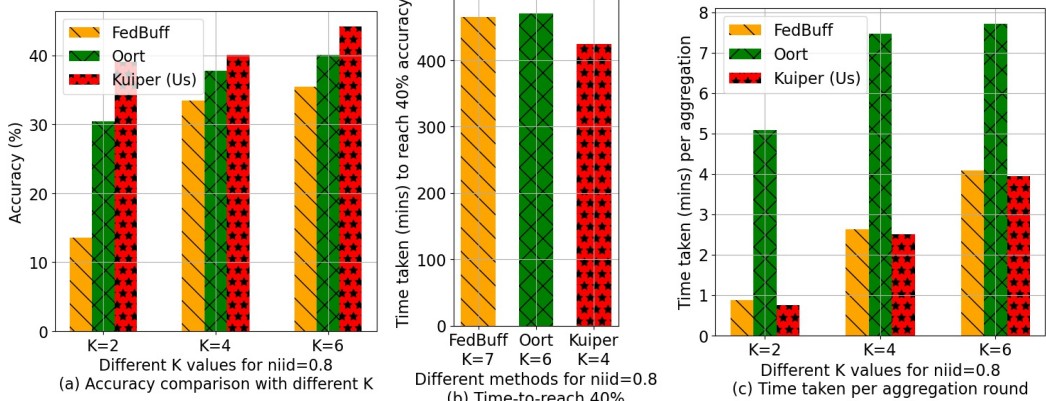

**Figure 6: Comparison of KUIPER with baselines Oort Lai et al. (2021) and FedBuff Nguyen et al. (2022).** (a) shows the effect of the parameter $K$ on the accuracy. As we increase $K$, it gets more synchronous, and thus accuracy increases, but the time taken for each round increases. (b) shows time taken to reach 40% accuracy. Here, Oort is using $K$=6 and FedBuff $K$=7 because they didn't achieve 40% accuracy with $K$=4. KUIPER is 11% faster than Oort and 10% faster than FedBuff. (c) Time taken per aggregation round. As we increase $K$, time taken for each aggregation round increases. Oort waits for specific $K$ clients but FedBuff and KUIPER aggregate the first $K$ clients thus taking less time per aggregation.

this we conclude that dropping large numbers of stragglers hurts performance. The overall results from (a) and (b) show the robustness of KUIPER in the presence of stragglers across the entire range of non-IIDness in data, without any significant drop in the validation accuracy and is not much below the ideal accuracy case of "Sync 12 homogeneous devices". All the other baselines degrade much faster than KUIPER, with increasing non-IID bias.

# 6 TAKEAWAYS

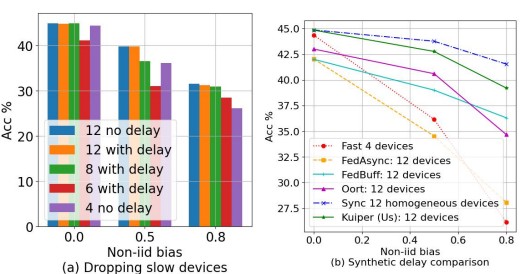

**Figure 7: Figures (a) and (b) show results on a 12 device setup: 4 devices with no delay, 4 with 3x delay, and 4 with 5x delay. Validation accuracy achieved when a straggler's updates are dropped as compared to (a) different delays (5x, 3x, no delay) and (b) weighing the stale updates in KUIPER. Highest validation accuracy is achieved by KUIPER because it considers updates from *all* clients. The improvement in accuracy increases with high non-IIDness.**

Here we have solved a heavyweight ML task, distributed action recognition, on edge devices with mobile GPUs using asynchronous FL as we have shown synchronous FL heavily suffers with the stragglers problem. We have considered a realistic scenario where all the clients might not have samples from each class (non-IID data distribution). Given the scarcity of the previous literature in asynchronous FL on heterogeneous and resource-constrained edge devices, we have proposed a new method called KUIPER, which is designed to handle both non-IID differences and heterogeneity in network speed and compute power of the clients. A unique design idea that we have developed in KUIPER is to consider both the speed of clients as well as the intrinsic value of the data at each client, when performing aggregation of gradient updates from each client. We have seen that a pure asynchronous approach does not work well and hence we have introduced a buffering strategy with a customizable "burst size" leading to a buffered asynchronous FL approach. We present a convergence proof of our approach, extending the analysis of FedBuff. Then, we empirically see that our KUIPER solution produces more accurate results on HMDB51 than the baselines (9% better than Oort, 12% than FedBuff, and 9% than FedAsync). For a comparable buffer size to reach the same accuracy, we are 11% and 10% faster than Oort and FedBuff respectively. Importantly, with hyperparameter tuning, we show that the per-clip accuracy achievable for buffered asynchronous federated learning (46.15%) is comparable to the case of a central server (47.8%) with no clients. Thus, we for the first time empirically show that it is possible to achieve activity recognition on edge devices that are already available for general release today. In future work, one may consider how to handle non-iid data in a personalized way to help cater to the specific needs of clients. One should also consider the effect of non-IIDness in feature space rather than just in classes.

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

## APPENDIX

## INTRODUCTION

In the supplementary material, we show the following:

- Link to our anonymized source code repository

- Algorithm

- Scalability of KUIPER: Performance with large number of devices ($M$=1000)

- Experiments on the UCF101 dataset with the 12-device setup

- Effect of number of clients and non-IID bias in final accuracy

- Effect of number of frames per clip in final accuracy

- Systematic delay experiment with the 4-device setup

- Knowledge distillation and effect of TA

- Convergence proof of KUIPER

## A  SOURCE CODE

We provide the source code of KUIPER at https://anonymous.4open.science/r/fedact_code-7513/). We have described in the README file how the edge devices can be prepared for running KUIPER.

## B ALGORITHM

---

**Algorithm 1** KUIPER

---

1: $T$: total iterations, $k \in [1, K]$: counter for clients reported in burst, $c_t$: set of clients in burst, $n_i$: samples present on $i^{th}$ device.
2: **Server**
3: Initialize $w_0$ from Knowledge Distillation model.
4: **for** $t = 1$ **to** $T$ **do**
5:     **if** Receive $(w^i_{new,t}, \tau^i)$ from any client **then**
6:         **if** $t < T_0$ **then**
7:             $w^{c_t}_{new,t} \leftarrow n_i \cdot w^i_{new,t} \times e(acctrain^i_t)$,
8:         **else**
9:             $w^{c_t}_{new,t} \leftarrow n_i \cdot w^i_{new,t}$
10:         **end if**
11:         $c_t \leftarrow c_t \cup \{i\}$
12:         $\tau^{c_t} \leftarrow \tau^{c_t} + \tau^i$
13:         $k \leftarrow k + 1$
14:         $N \leftarrow N + n_i$
15:     **end if**
16:     **if** $k == K$ **then**
17:         $w^{c_t}_{new,t} \leftarrow \frac{w^{c_t}_{new,t}}{N}$
18:         $\tau^{c_t} \leftarrow \frac{\tau^{c_t}}{K}$
19:         $\beta^{c_t}_t \leftarrow \beta \times s(t - \tau^{c_t})$ , $s(\cdot)$ is a function of the staleness
20:         $w^g_t \leftarrow (1 - \beta^{c_t}_t) w^g_{t-1} + \beta^{c_t}_t w^k_{new}$
21:         send $w^g_t$ to $client^j \; \forall j \in c_t$
22:         $w^{c_t}_{new,t} \leftarrow 0, k \leftarrow 0, N \leftarrow 0, c_t \leftarrow \{\}$
23:     **end if**
24: **end for**
25: **Client**
26: **for** $i \in \{1, \ldots, n\}$ in parallel **do**
27:     receive global model and time stamp $(w^g_t, t)$
28:     $\tau \leftarrow t$ , $w \leftarrow w^g_t$
29:     **for** local iteration $h = 1 : H$ **do**
30:         $w_h \leftarrow w_{h-1} - \eta \nabla g_{w_h}$
31:     **end for**
32:     Send $(w_H, \tau)$ to the server
33: **end for**

---

## C UCF101 EXPERIMENT FOR 12-DEVICE SETUP

Figure 8 shows the 12-device experiment on UCF101 dataset with $K = 4$ complementing the experiment (shown in Fig. 3(b) of the main paper with the HMDB51 dataset). This experiment compares KUIPER with the other three baselines - FedAsync, FedBuff, and Oort. As KUIPER appropriately balances the delay penalty and data quality reward, it performs better than the other baselines. It is also interesting to observe that Oort takes more time than FedBuff when non-iid bias is zero. This is because it waits for specific clients according to their statistical utility and all the clients have the same data distribution in an iid setting. However, when the non-iid bias is 0.8, Oort takes less time than FedBuff to reach 25% accuracy as selecting the specific clients with more useful data helps to aggregate a better global model (Figure 8 (b)).

## D KUIPER COMPONENTS AND HYPERPARAMETERS

Figure 9(a) shows how each component is KUIPER affects the accuracy. When non-iid=0 (iid case), the error reward term is not improving any accuracy as the data among all the clients is IID and the model can equally learn from any client. Figure 9(b, c) shows the effect of staleness penalty $\alpha$ and mixing hyperparameter $\beta$. For higher non-IID bias, increasing $\alpha$ reduces accuracy drastically (4% for non-IID=0.5 and 8% for non-IID=0.8); when we change $\alpha$ from 0.7 to 1.0, slow clients have exclusive data, and global model can learn even from the stale gradients. $\beta$ controls how much the global model should change with the new model updates. We find expectedly that too slow a change as well as too fast a change hurts accuracy.

## E EFFECT OF NON-IID BIAS AND NUMBER OF CLIENTS

Figure 10 shows how accuracy changes with different numbers of clients and non-IID bias. We have shown this analysis for three image recognition datasets (MNIST, FMNIST, and CIFAR10). For a high number of clients, increasing non-IID bias results in a drastic decrease in the accuracy compared to the case where the number of clients is low.

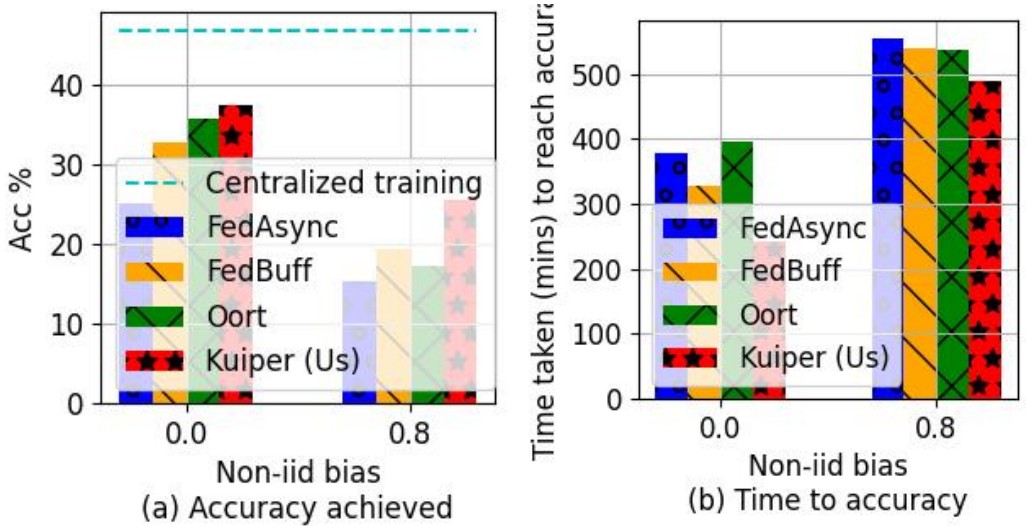

**Figure 8: Comparison of KUIPER with the FedAsync, FedBuff, and Oort baselines for 12 device setup for UCF101 dataset (a) Accuracy achieved with different non-iid bias values and (b) Time taken to reach accuracy of 25% when non-iid bias is 0 and 15% when non-iid bias is 0.8. KUIPER achieves higher accuracy and takes less time to reach the accuracy than the other baselines with a low and high non-iid value.**

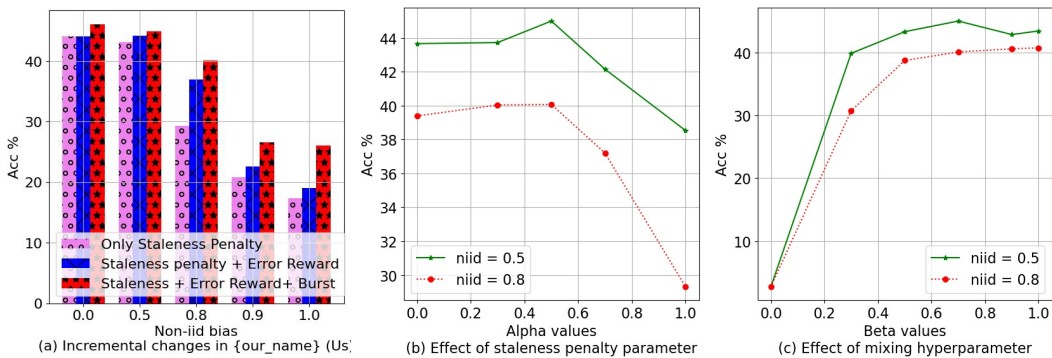

**Figure 9: (a) Performance improvement due to each of the three design elements of KUIPER, (b) Effect of $\alpha$: As non-IID factor increases (0.5 to 0.8), penalizing from $\alpha$=0.7 to $\alpha$=1.0 shows significant drop in accuracy, (c) Effect of $\beta$ parameter with varying non-iid bias. When $\beta$=0. the model is stuck with the initial weights and prediction is random. Too high a $\beta$ causes a drop in accuracy.**

## F    EFFECT OF NUMBER OF FRAMES PER CLIP IN FINAL ACCURACY

Figure 11 shows how accuracy changes with the number of frames. With more frames per video, it is easier for the model to recognize the action, and thus accuracy increases as the number of frames per video increases. Due to limited computational resources, we cannot increase the number of frames and thus fix it to 8 frames for the HMDB51 dataset and 32 for the UCF101.

## G    EFFECT OF STRAGGLERS: DELAY EXPERIMENT

Figure 12 shows how the inclusion of heterogeneous devices changes the training trajectory. Figure 12 (a, b) compares the two setups where all four devices are homogeneous (NX)) vs. three devices are the same (NX), and one slow device (Nano) is there. Updates of this device are delayed and thus stale. With a slow device, accuracy decreases (Figure 12 (a)), and the time taken to reach a specific accuracy increases. Here, note that Oort waits explicitly for the $K$ clients depending on their utility

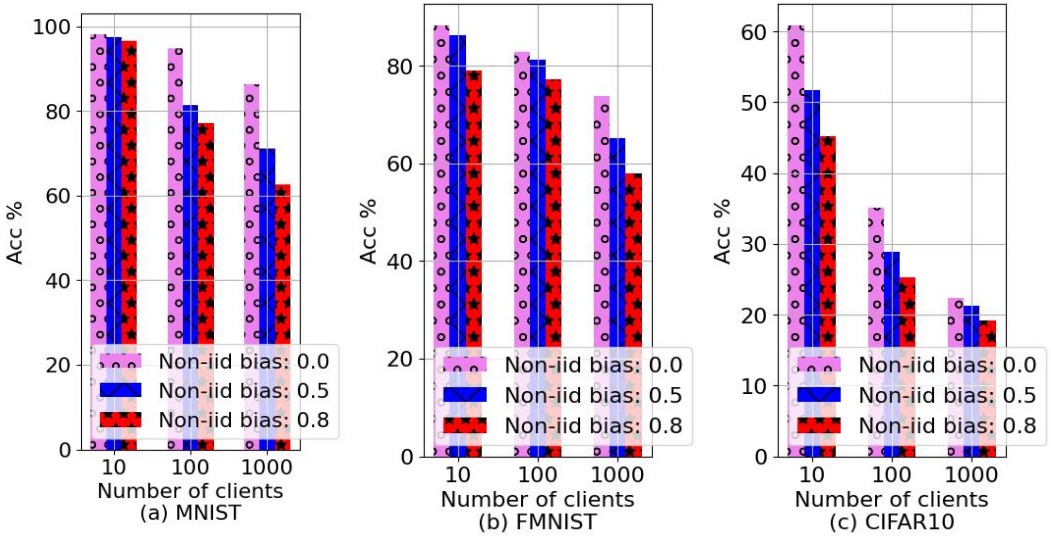

**Figure 10: Changing accuracy with changing number of clients and Non-IID bias for (a) MNIST (b) FMNIST (c) CIFAR10**

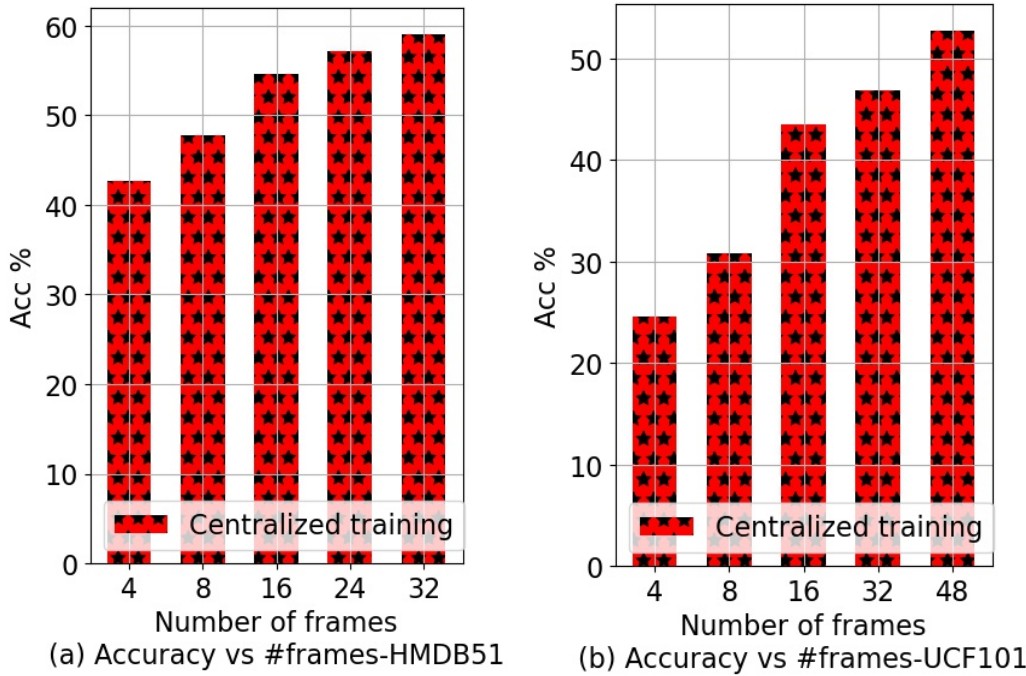

**Figure 11: Centralized training with different number of frames for (a) HMDB51 and (b) UCF101. Accuracy increases as the number of frames per video increases.**

scores, and waiting for the delayed client makes Oort slower than FedBuff even though it was faster in the homogeneous case

Figure 13 shows the effect of delay in a 4-device setup, similar to Fig.5(c, d) in the main paper. In this experiment, 1x, 3x, 5x delays are a multiple of natural delay. The delay ratios have been chosen in order to mimic a realistic scenario, as described in "Effect of stragglers" in **Experimental Evaluation** section of the main paper. Here, the accuracy when the slowest device is dropped, is comparable to others for non-iid values of 0 and 0.5, but it gets drastically reduced for 0.8 non-iid value. Because

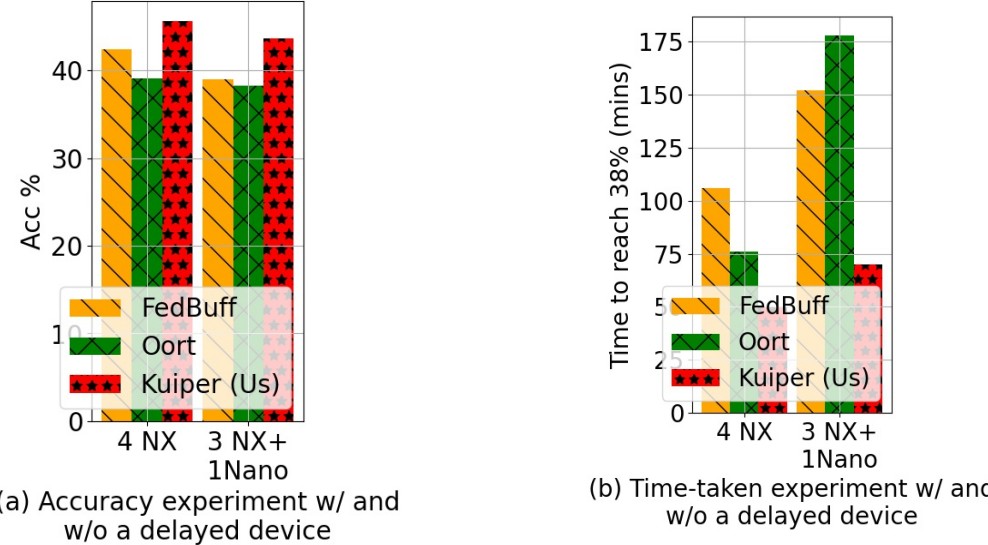

**Figure 12: Comparison when all four devices are homogeneous vs. three devices are homogeneous and one device is slow. (a) Accuracy achieved with three different methods (b) Time taken to reach 38% accuracy in two setups.**

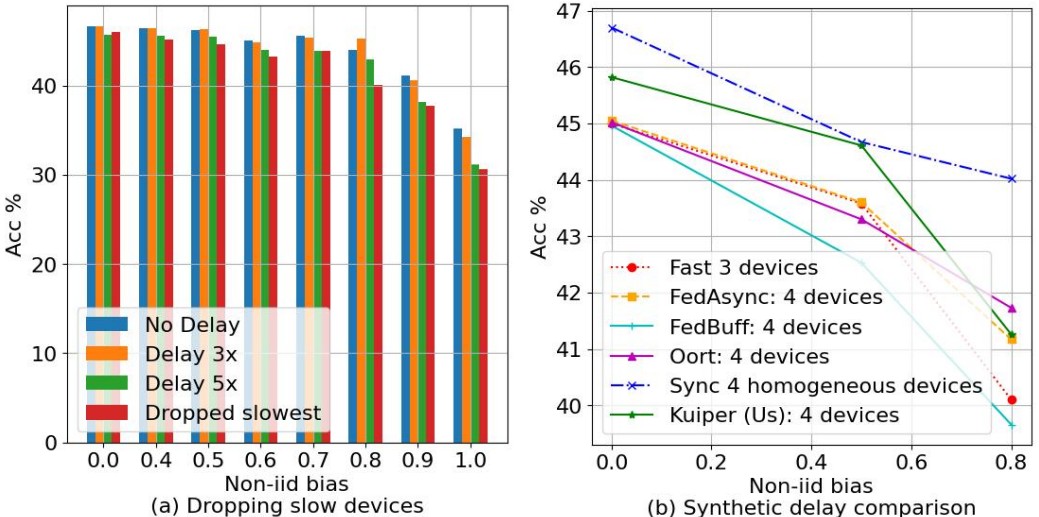

**Figure 13: Four device setup: one devices with no delay, one with 1x delay, one with 3x delay, and one with 5x delay. Validation accuracy achieved when a straggler's updates are dropped as compared to (a) different delays (5x, 3x, 1x, no delay) and (b) weighing the stale updates in KUIPER. Highest validation accuracy is achieved by KUIPER because it considers updates from *all* clients. The improvement in accuracy increases with high non-iidness.**

with high non-iid value, clients get more data of some specific classes and losing that client means losing the training data for that class (Figure 13(b)).

## H    KNOWLEDGE DISTILLATION

In the first stage of our pipeline, we perform knowledge distillation from a larger model, trained on the Kinetics dataset. We compare three approaches in order to validate using knowledge distillation with an intermediate TA. For these experiments, we use a batch size of 128, learning rate $\eta = 0.1$, and an SGD optimizer with a weight decay 0.001 and momentum 0.9. In the first experiment, we train a ResNet-18 model from scratch on the Kinetics dataset and the per-clip top-1 accuracy achieved is

Table 1: ResNet-18 training time per epoch. For HMDB51 and UCF101, each client has approximately 500MB and 1.73GB of video data respectively

| Dataset | Device | Train Time (per local epoch) |
|---------|--------|------------------------------|
| HMDB51 | NVIDIA Jetson Nano | 391.1 seconds |
| HMDB51 | NVIDIA Jetson TX2 | 293.1 seconds |
| HMDB51 | NVIDIA Jetson Xavier NX | 121.3 seconds |
| HMDB51 | NVIDIA Jetson AGX Xavier | 84.5 seconds |
| UCF101 | NVIDIA Jetson Nano | 2691.6 seconds |
| UCF101 | NVIDIA Jetson TX2 | 2001.4 seconds |
| UCF101 | NVIDIA Jetson Xavier NX | 821.9 seconds |
| UCF101 | NVIDIA Jetson AGX Xavier | 572.1 seconds |

Table 2: ResNet-18 evaluated on the entire test dataset. The device heterogeneity is reflected in the inference times

| Dataset | Device | Test Time |
|---------|--------|-----------|
| HMDB51 | NVIDIA Jetson Nano | 181.4 seconds |
| HMDB51 | NVIDIA Jetson TX2 | 116.3 seconds |
| HMDB51 | NVIDIA Jetson Xavier NX | 89.4 seconds |
| HMDB51 | NVIDIA Jetson AGX Xavier | 68.3 seconds |
| UCF101 | NVIDIA Jetson Nano | 621.3 seconds |
| UCF101 | NVIDIA Jetson TX2 | 381.2 seconds |
| UCF101 | NVIDIA Jetson Xavier NX | 322.5 seconds |
| UCF101 | NVIDIA Jetson AGX Xavier | 217.7 seconds |

50.2%. Using knowledge distillation, the accuracy is improved to 53.8% when we distill directly from ResNet-34 to ResNet-18, and 54.6% when we use a ResNet-26 as the intermediate TA between the teacher and student. From Figure 14, it is evident that using a distilled ResNet-18 is better than using a ResNet-18 trained from scratch. There is a counter pull from the training time — the KD approach (discounting the time to train the ResNet-34) takes 43% longer than training from scratch (ResNet-18). This can be explained by the fact that "Train from scratch" includes only forward-backward passes on ResNet-18 with optimization using only cross-entropy loss. On the other hand, KD involves forward passes on the larger ResNet-34, forward-backward passes on ResNet-18, and optimization on ResNet-18 using a combination of both cross-entropy loss and the MSE on the logits (recall that we are fine-tuning only the last FC layer). This timing result is consistent with prior works that report on the timing performance of knowledge distillation Hinton et al. (2015); Sun et al. (2019).

We further investigate using multiple TAs. From Table 4, we see that the introduction of one TA increases the train time from 44 hours 58 minutes to 55 hours 23 minutes and the corresponding increase in per-clip accuracy is 0.8%. Hence, there is a trade-off between increased training time and increased accuracy. Furthermore, the introduction of a TA almost always increases accuracy but the optimal number of TAs and size of each is an open research question Mirzadeh et al. (2020). Additionally, TAs are used to bridge the gap between the student and teacher: by using a ResNet-26 between ResNet-34 and ResNet-18 we already accomplish this. If the gap between the teacher and student were larger, using additional TAs would be of benefit at the expense of increased computation and train time required. In order to reduce the train times and achieve comparable accuracy to the baselines, we use one TA.

From Figure 14, we conclude that using KD does give an improvement to all FL algorithms, with the improvement being most significant for FedAsync (a 6% increase). KUIPER enjoys an improvement of 3%.

We further investigate the effects of using additional TAs in our pipeline.

In all these experiments, distillation is performed from teacher ResNet-34 to student ResNet-18. In the first experiment, we do not use any TAs. In the next experiment, we use ResNet-26 as a TA. The

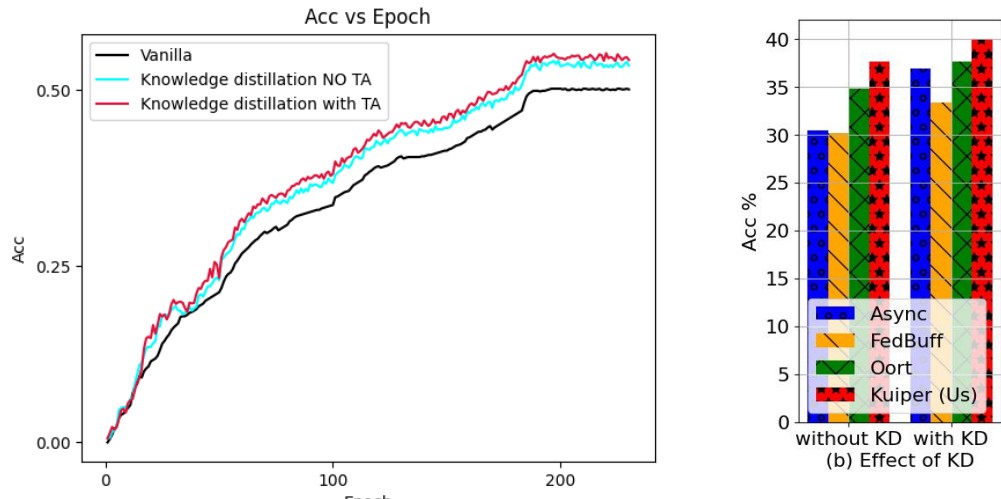

Figure 14: (a) Accuracy on the Kinetics validation dataset for 3 experiments: 1. Training a ResNet-18 from scratch (Vanilla); 2. Knowledge distillation from ResNet-34 to ResNet-18 (Knowledge Distillation with no TA); 3. Knowledge distillation with ResNet-34 Teacher, ResNet-26 TA, and ResNet-18 Student (Knowledge Distillation with TA) (b) Ablation study to show the importance of KD (Knowledge Distillation).

Table 3: Knowledge distillation is performed by varying the number of intermediate Teaching Assistants (TAs)

| # TAs | Epochs | Time (hrs, mins) (Increase) | Per-Clip Accuracy |
|-------|--------|------------------------------|-------------------|
| 0 | 200 | 44 h 58 m (0%) | 53.8% |
| 1 | 200 | 55 h 23 m (23.2%) | 54.6% |
| 2 | 200 | 69 h 35 m (54.7%) | 54.8% |
| 3 | 200 | 85 h 47 m (90.8%) | 54.9% |

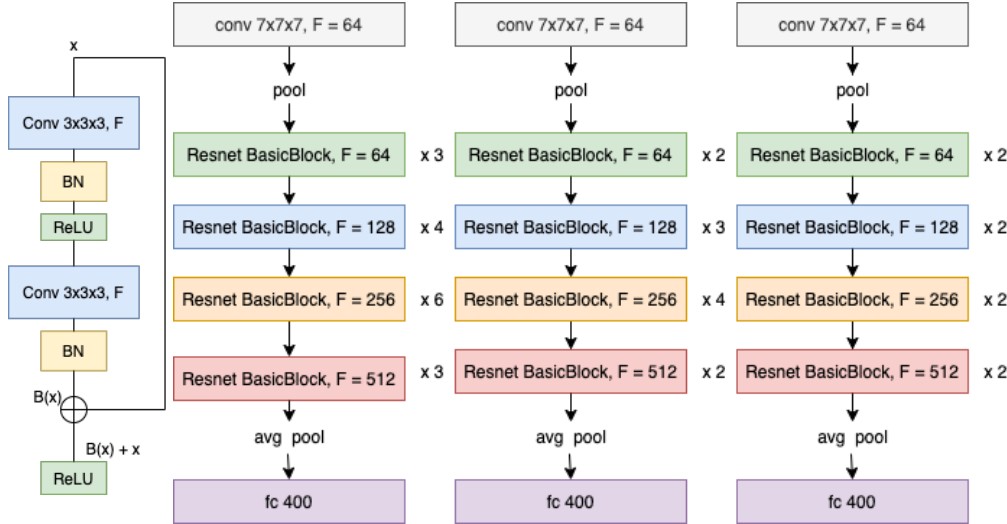

Figure 15: (a) The central server first performs knowledge distillation using the Kinetics dataset: from teacher to teaching assistant (TA) and from TA to student. The students (compressed models) are then fine-tuned on the smaller dataset using an asynchronous federated optimization. (b) ResNet-34, ResNet-26, and ResNet-18 architectures derived from the basic building block.

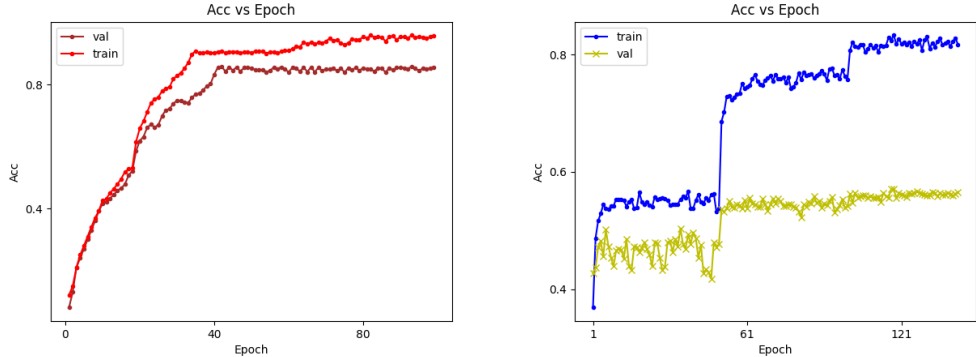

Figure 16: Central server: (a) ResNet-18, distilled from ResNet-34, via a ResNet-26 Teaching assistant, and fine-tuned on UCF101, performed at the central server without any clients (b) ResNet-18, distilled from ResNet-34, via a ResNet-26 Teaching assistant, and fine-tuned on HMDB51. The fine-tuning is performed at the central server without any clients

third experiment is performed using two TAs: ResNet-28 and ResNet-24, and in the final experiment, we use three TAs: ResNet-30, ResNet-26, and ResNet-22. The architectures have been outlined in Figure 15. From Table 3, we see that while the increase in **per-clip top-1** accuracy is appreciable when one TA is introduced, using additional TAs does not produce any considerable improvement in accuracy. The training time increases sharply as more TAs are added. Hence, in the subsequent stages in our pipeline, we chose to use a single TA.

For the rest of the experiments, we perform fine-tuning, by reinitializing the fully connected layer — the last layer in the ResNet-18 model. The ResNet-18 being used is the model distilled from ResNet-34 (trained on the Kinetics dataset) via a ResNet-26 TA.

**Datasets** The Kinetics-400 dataset requires an approximate disk space of 400GB to store. Amongst the edge devices we are using in these experiments, the most well-endowed, NVIDIA Jetson AGX Xavier has only 32GB storage. Hence, edge devices can only accommodate smaller-sized datasets on

**Table 4:** The time required for various stages shown in Figure 15 (a) and the baseline experiments. KD refers to Knowledge Distillation, synchronous refers to fine-tuning using FedAvg, asynchronous refers to fine-tuning using asynchronous federated optimization. The fine-tuning is performed using ResNet-18, distilled from ResNet-34 via a ResNet-26 Teaching assistant

| Dataset | Task | Epochs | Time |
|---------|------|--------|------|
| Kinetics | Train from scratch | 200 | 31 hrs 26 mins |
| Kinetics | KD (No TA) | 200 | 44 hrs 58 mins |
| Kinetics | KD (TA) | 200 | 55 hrs 23 mins |
| HMDB51 | Fine-tune no clients | 80 | 3hrs 15mins |
| HMDB51 | Synchronous | 80 | 10 hrs 54 mins |
| HMDB51 | Asynchronous | 80 | 6 hrs 31 mins |
| UCF101 | Fine-tune no clients | 80 | 22 hrs 5 mins |
| UCF101 | Synchronous | 80 | 74 hrs 27 mins |
| UCf101 | Asynchronous | 80 | 44 hrs 7 mins |

them[4]. In this section, we use the HMDB51 dataset and the UCF101 for evaluation. The HMDB51 which has a size of 2,062MB and is distributed amongst the clients in such a way that requires approximately 500MB of storage space on each client. The UCF101 is 6.9GB and each client has about 1.725GB of data. Once we have distilled knowledge from the larger ResNet-34, trained on the Kinetics dataset to the ResNet-18 architecture via a TA, the next step is to fine-tune on the smaller dataset; i.e., HMDB51 or UCF101.

From Table 4, refer to the HMDB51 experiments. We see that the time required for a synchronous optimization is 10 hours and 54 minutes. In contrast, the asynchronous federated algorithm takes only 6 hours and 31 minutes, a 40% decrease. This can be attributed to the clients having different computing resources, and hence requiring different amounts of time to complete the local epochs as given in Table 1. While the synchronous algorithm has to wait for the slowest client to send its update, the asynchronous algorithm continues its optimization. A similar effect is observed in the case of UCF101.

One may wonder that it is beneficial to use the approach of fine tuning at the server without any clients (for HMDB51 and UCF101) and thus not having to use our approach. This alternate method runs into the problem that it does not leverage federated learning, which has its traditional benefits of scaling to a large number of clients (and thus not needing heavyweight server) and preserving privacy of client data. The same argument applies to why we would not want to train for the Kinetics data from scratch (this would obviously have to be done at the server).

## I CONVERGENCE PROOF

Here we provide the complete convergence analysis of KUIPER. This analysis is influenced by FedBuff Nguyen et al. (2022) and is customized to our model. Specifically, we have characterized the effect of non-iid bias on gradient variances and the convergence guarantee.

**Notation.** M denotes total number of clients. $g_i(w; \zeta_i)$ denotes stochastic gradient on $i^{th}$ client on a model with weights $w$ and sampled batch $\zeta_i$. $\nabla F_i(w)$ denotes the gradient with respect to the loss. $\sigma_l^2$ and $\sigma_g^2$ are local and global variances of the gradients. $f(w)$ is the objective function as described below, and $f^*$ is its theoretical minima. $t$ is the current iteration count and $\tau_i$ is the global iteration count when $i^{th}$ client received gradients from the server where every client runs $Q$ local iterations before communicating with the server.
The objective function is formally defined as

$$\min_{w \in \mathbb{R}^d} f(w) := \frac{1}{m} \sum_{i=1}^{m} p_i F_i(w) \tag{7}$$

---

[4]One may argue that adding cheap external storage such as through Flash cards can alleviate this problem. However, reading from external storage is orders of magnitude slower than reading from internal storage and will thus increase the training time to an infeasible level.

where $p_i$ is the weight assigned to the updates coming from client $i$. We make the following assumptions for proving the convergence of KUIPER. Assumptions 1-4 are standard assumptions made for the convergence analysis of any synchronous FL system. Assumption 5 pertains to an asynchronous FL system.

**Assumption 1:** (Unbiased client stochastic gradients) $\mathbb{E}[g_i(w; \zeta_i)] = \nabla F_i(w)$.

**Assumption 2:** (Bounded local and global variance) $\forall i \in [M]$, $\mathbb{E}_{\zeta_i|i}[\|g_i(w; \zeta_i) - \nabla F_i(w)\|^2] \leq \sigma_l^2$ and $\frac{1}{M} \sum_{i=1}^{m} \|\nabla F_i(w) - \nabla f(w)\|^2 \leq \sigma_g^2$.

**Assumption 3:** (Gradients are bounded) $\|\nabla F_i\|^2 \leq G$.

**Assumption 4:** (L-smoothness), $\forall i \in [M]$, the gradient is L-smooth, $\|\nabla F_i(w) - \nabla F_i(w')\|^2 \leq L\|w - w'\|^2$.

**Assumption 5:** (Bounded Staleness) The staleness of stragglers $t - \tau$, where t represents current global epoch and $\tau$ represents the global epoch when the client last synchronized with the server, is bounded $t - \tau \leq \tau_{max,1}$ which is the maximum across all the clients.

**Theorem 1-** *Let $\eta_l^{(q)}$ be the local learning rate of client SGD in the q-th step, and define $\alpha_1(Q) := \sum_{q=0}^{Q-1} \eta_l^{(q)}$ and $\alpha_2(Q) := \sum_{q=0}^{Q-1} (\eta_l^{(q)})^2$. Choosing $\eta_g \eta_l^{(q)} Q \leq \frac{1}{L}$ for all local steps $q = 0, ..., Q-1$, the global model iterates in Algorithm 1 achieves the following ergodic convergence rate*

$$\frac{1}{T} \sum_{t=0}^{T-1} \|\nabla_f(w^t)\|^2 \leqslant \frac{2(f(w^0) - f^*)}{\eta_g \alpha_1(Q)T} + \frac{L}{2} \frac{\eta_g \alpha_2(Q)}{\alpha_1(Q)} \sigma_l^2$$
$$+ 3L^2 Q \alpha_2(Q) (\eta_g^2 \tau_{max,K}^2 + 1)(\sigma_l^2 + \sigma_g^2 + G) \tag{8}$$

**Corollary 1-** *Choosing a constant local learning rate $\eta_l$ and global learning rate $\eta_g$ such that $\eta_g \eta_l Q \leq \frac{1}{L}$, the global model iterates in KUIPER are bounded by*

$$\frac{1}{T} \sum_{t=0}^{T-1} \mathbb{E}[\|\nabla f(w^t)\|^2] \leq \frac{2F^*}{\eta_g \eta_l QT} + \frac{L}{2} \eta_g \eta_l \sigma_l'^2 + 3L^2 Q^2 \eta_l^2 (\eta_g^2 \tau_{max,K}^2 + 1)\sigma'^2 \tag{9}$$

where $F^* := f(w^0) - f^*$, $\sigma^2 := \sigma_l'^2 + \sigma_g'^2 + G'$. In KUIPER, we rescale the client's gradients by $e(\cdot)$, which is bounded by 0 at the lower and 1 at the higher end. Specifically, $\nabla F_i'(w) = \nabla F_i(w) \times e(error_i)$, thus introducing $\sigma_l'$, $\sigma_g'$, and $G'$ as the new bounds of local variance, global variance, and norm of gradients when the gradient updates respectively. $Q$ is the number of local iterations for a client, and $T$ the total number of global iterations. Further, choosing $\eta_l = \mathcal{O}(1/(K\sqrt{TQ}))$ and $\eta_g = \mathcal{O}(K)$, for all $\eta_g, \eta_l$ satisfying $\eta_g \eta_l Q \leq \frac{1}{L}$ and sufficiently large $T$, we have

$$\frac{1}{T} \sum_{t=0}^{T-1} \|\nabla f(w^t)\|^2 \leq \mathcal{O}(\frac{F^*}{\sqrt{TQ}}) + \mathcal{O}(\frac{\sigma_l'^2}{\sqrt{TQ}}) + \mathcal{O}(\frac{Q\sigma'^2}{TK^2}) + \mathcal{O}(\frac{Q\sigma'^2 \tau_{max,1}^2}{TK^2}) \tag{10}$$

Reacall tha, in KUIPER, we rescale $\eta_g$ with $s(\cdot)$ that is bounded by 0 at the lower and 1 at the upper end. Specifically, $\beta = \eta_g \times s(t - \tau)$. The modified global rate $\beta$ is therefore still bounded by $\mathcal{O}(K)$ and satisfies the above results. For sufficiently large $T$, the algorithm achieves the convergence rate as shown in Eq. (10). As we can see, convergence guarantee increases with increasing $K$ as we tend to go closer to the synchronous aggregation. Also, as non-iid bias increases, gradient variances increase, and weakens the convergence guarantee. Having described how the two modifications in KUIPER do not affect further analysis, we describe the rest of the formal proof without the two modifications for simplicity.

We now state a useful Lemma that will help us prove the above theorem.

**Lemma 1-** $\mathbb{E}\left[\|g_k\|^2\right] \leqslant 3(\sigma_l^2 + \sigma_g^2 + G_1)$, where the total expectation $\mathbb{E}[\cdot]$ is evaluated over the randomness with respect to client participation and the stochastic gradient taken by a client.

**Proof of Lemma 1-** From the law of total expectation we have $\mathbb{E} = \mathbb{E}_{k \sim [m]} \mathbb{E}_{\zeta_k|k}$. Hence,

$$\mathbb{E}\left[\|g_k(w)\|^2\right] = \mathbb{E}_{k \sim [m]} \mathbb{F}_{g|k}\left[\|g_k(w) - \nabla F_k(w) + \nabla F_k(w) - \nabla f(w) + \nabla f(w)\|^2\right]$$
$$\leq 3\mathbb{E}_{k \sim [m]} \mathbb{E}_{g|k}\left[\|g_k(w) - \nabla F_k(w)\|^2 + \|\nabla F_k(w) - \nabla f(w)\|^2 + \|\nabla f(w)\|^2\right]$$
$$= 3(\sigma_l^2 + \sigma_g^2 + G) \tag{11}$$

We now define Theorem 2 which we will use to prove Theorem 1.

**Theorem 2-**
*Let $\eta_l^{(q)}$ be the local learning rate of client SGD in the q-th step, and define $\alpha_1(Q) := \sum_{q=0}^{Q-1} \eta_l^{(q)}$,*
*$\alpha_2(Q) := \sum_{q=0}^{Q-1} (\eta_l^{(q)})^2$. Choosing $\eta_g \eta_l^{(q)} Q \leq \frac{1}{L}$ for all local steps $q = 0, .., Q-1$, the global*
*model iterates in Algorithm 1 achieves the following ergodic convergence rate*

$$\frac{1}{T} \sum_{t=0}^{T-1} \|\nabla f(w^t)\|^2 \leq \frac{2(f(w^0) - f(w^*))}{\eta_g \alpha_1(Q) T} + 3L^2 Q \alpha_2(Q)(\eta_g^2 \tau_{max,K}^2 + 1)(\sigma_l^2 + \sigma_g^2 + G) + \frac{L}{2} \frac{\eta_g \alpha_2(Q)}{\alpha_1(Q)} \sigma_l^2.$$
(12)

**Proof of Theorem 2-** By $L$-smoothness assumption,

$$f\left(w^{t+1}\right) \leq f\left(w^t\right) - \eta_g \langle \nabla f\left(w^t\right), \overline{\Delta}^t \rangle + L \frac{\eta_g^2}{2} \|\overline{\Delta}^t\|^2$$

$$\leq f\left(w^t\right) - \underbrace{\frac{\eta_g}{K} \sum_{k \in S_t} \langle \nabla f\left(w^t\right), \Delta_k^{t-\tau_k} \rangle}_{T_1} + \underbrace{\frac{L\eta_g^2}{2K^2} \| \sum_{k \in S_t} \|\Delta_k^{t-\tau_k}\|^2}_{T_2}$$
(13)

where $\Delta_k^{t-\tau_k}$ is the client delta which is trained from using the global model after $t - \tau_k$ updates as initialization. We will next derive the upper bounds on $T_1$ and $T_2$. To begin,

$$T_1 = -\frac{\eta_g}{K} \sum_{k \in S_t} \left\langle \nabla f\left(w^t\right), \sum_{q=0}^{Q-1} \eta_l^{(q)} g_k\left(y_{k,q}^{t-\tau_k}\right) \right\rangle$$

$$= -\frac{\eta_g}{K} \sum_{k \in S_t} \sum_{q=0}^{Q-1} \eta_l^{(q)} \left\langle \nabla f\left(w^t\right), g_k\left(y_{k,q}^{t-\tau_k}\right) \right\rangle$$
(14)

Using conditional expectation, the expectation operator can be written as

$$\mathbb{E}[\cdot] := \mathbb{E}_{\mathcal{H}} \mathbb{E}_{i \sim [m]} \mathbb{E}_{g_i | i, \mathcal{H}}[\cdot]$$

where $\mathbb{E}_{\mathcal{H}}$ is the expectation over the history of the iterates, $\mathbb{E}_{i \sim [M]}$ is evaluated over the randomness over the distribution of clients $i \sim [M]$ checking in at time-step $t$, and the inner expectation operates over the stochastic gradient of one step on a client. Hence, following unbiasedness,

$$\mathbb{E}\left[T_1\right] = -\mathbb{E}\left[\frac{\eta_g}{K} \sum_{k \in S_t} \sum_{q=0}^{Q-1} \eta_l^{(q)} \left\langle \nabla f\left(w^t\right), g_k\left(y_{k,q}^{t-\tau_k}\right) \right\rangle\right]$$

$$= -\eta_g \mathbb{E}_{\mathcal{H}}\left[\frac{1}{m} \sum_{i=0}^{m} \sum_{q=0}^{Q-1} \eta_l^{(q)} \mathbb{E}_{g_i | i \sim [m]} \left\langle \nabla f\left(w^t\right), g_i\left(y_{i,q}^{t-\tau_i}\right) \right\rangle\right]$$

$$= -\frac{\eta_g}{m} \mathbb{E}_{\mathcal{H}}\left[\sum_{i=0}^{m} \sum_{q=0}^{Q-1} \eta_l^{(q)} \left\langle \nabla f\left(w^t\right), \nabla F_i\left(y_{i,q}^{t-\tau_i}\right) \right\rangle\right]$$

$$= -\frac{\eta_g}{m} \mathbb{E}_{\mathcal{H}}\left[\sum_{q=0}^{Q-1} \eta_l^{(q)} \left\langle \nabla f\left(w^t\right), \frac{1}{m} \sum_{i=0}^{m} \nabla F_i\left(y_{i,q}^{t-\tau_i}\right) \right\rangle\right]$$
(15)

From the identity,

$$\langle a, b \rangle = \frac{1}{2}\left(\|a\|^2 + \|b\|^2 - \|a - b\|^2\right)$$

we have

$$\mathbb{E}[T_1] = \frac{-\eta_g}{2} \left( \sum_{q=0}^{Q-1} \eta_l^{(q)} \right) \left\| \nabla f \left( w^t \right) \right\|^2 + \sum_{q=0}^{Q-1} \frac{\eta_g \eta_l^{(q)}}{2} \left( -\mathbb{E}_{\mathcal{H}} \left\| \frac{1}{m} \sum_{i=1}^{m} \nabla F_i \left( y_{i,q}^{t-\tau_i} \right) \right\| \right.$$

$$\left. + \underbrace{\mathbb{E}_{\mathcal{H}} \left\| \nabla f \left( w^t \right) - \frac{1}{m} \sum_{i=1}^{m} \nabla F_i \left( y_{i,q}^{t-\tau_i} \right) \right\|^2}_{T_3} \right) \tag{16}$$

Now for $T_3$, from the definition of $f(w^t)$,

$$\mathbb{E}_{\mathcal{H}}[T_3] = \mathbb{E}_{\mathcal{H}} \left\| \frac{1}{m} \sum_{i=1}^{m} \nabla F_i \left( w^t \right) - \frac{1}{m} \sum_{i=1}^{m} \nabla F_i \left( y_{i,q}^{t-\tau_i} \right) \right\|^2$$

$$\leq \frac{1}{m} \sum_{i=1}^{m} \mathbb{E}_{\mathcal{H}} \left\| \nabla F_i \left( w^t \right) - \nabla F_i \left( y_{i,q}^{t-\tau_i}; \right) \right\|^2 \tag{17}$$

Further, by telescoping, $T_3$ can be decomposed as

$$\mathbb{E}[T_3] = \frac{1}{m} \sum_{i=1}^{m} \mathbb{E}_{\mathcal{H}} \left\| \nabla F_i \left( w^t \right) - \nabla F_i \left( w^{t-\tau_i} \right) + \nabla F_i \left( w^{t-\tau_i} \right) - \nabla F_i \left( y_{i,q}^{t-\tau_i} \right) \right\|^2$$

$$\leqslant \frac{2}{m} \sum_{i=1}^{m} \mathbb{E}_{\mathcal{H}} \left( \underbrace{\left\| \nabla F_i \left( w^t \right) - \nabla F_i \left( w^{t-\tau_i} \right) \right\|^2}_{\text{staleness}} + \underbrace{\left\| \nabla F_i \left( w^{t-\tau_i} \right) - \nabla F_i \left( y_{i,q}^{t-\tau_i} \right) \right\|^2}_{\text{local drift}} \right) \tag{18}$$

$$\leqslant \frac{2}{m} \sum_{i=1}^{m} \left( L^2 \mathbb{E}_{\mathcal{H}} \left\| w^t - w^{t-\tau_i} \right\|^2 + L^2 \mathbb{E}_{\mathcal{H}} \left\| w^{t-\tau_i} - y_{i,q}^{t-\tau_i} \right\|^2 \right)$$

The upper bound on $T_3$ can be understood as sums of bounds on the effect of staleness and local drift during client training, and local variance induced by client-side SGD. Further, we need to produce an upper bound on the staleness of initial model from which the client models are trained.

$$\left\| w^t - w^{t-\tau_i} \right\|^2 = \left\| \sum_{\rho=t-\tau_i}^{i-1} \left( w^{\rho+1} - w_{\rho} \right) \right\|^2 = \left\| \sum_{\rho=t-\tau_i}^{t-1} \frac{\eta_g}{K} \sum_{j_\rho \in S_\rho} \Delta_{j_\rho}^{\rho} \right\|^2$$

$$= \frac{\eta_g^2}{K^2} \left\| \sum_{\rho=t-\tau_i}^{t-1} \sum_{j_\rho \in S_\rho} \sum_{l=0}^{Q-1} \eta_l^{(l)} g_{j_\rho} \left( y_{j_\rho,l}^{\rho} \right) \right\|^2 \tag{19}$$

Taking the expectation in terms of $\mathcal{H}$,

$$\left| E_{\mathcal{H}} \left\| w^t - w^{t-\tau_i} \right\|^2 \leq \frac{\eta_g^2 Q \tau_i}{K} \sum_{\rho=t-\tau_i}^{t-1} \sum_{j_\rho \in S_\rho} \sum_{l=0}^{Q-1} (\eta_l^{(l)})^2 \mathbb{E} \| g_{j_\rho} (y_{j_\rho,l}^{\rho}) \|^2 \right.$$

$$\leqslant 3\eta_g^2 Q \max_{\tau_i} \tau_i^2 \left( \sum_{l=0}^{Q-1} \left( \eta_l^{(l)} \right)^2 \right) \left( \sigma_1^2 + \sigma_g^2 + G \right)$$

$$\leqslant 3\eta_g^2 Q \tau_{max,K}^2 \left( \sum_{l=0}^{Q-1} \left( \eta_l^{(l)} \right)^2 \right) \left( \sigma_1^2 + \sigma_g^2 + G \right) \tag{20}$$

where the last inequality follows from the assumption on maximal delay and applying Lemma 1 (Eqn. 11). Similarly, the local drift term can be upper-bounded by

$$\mathbb{E} \left\| w^{t-\tau_i} - y_{i,q}^{t-\tau_i} \right\|^2 = \mathbb{E} \left\| y_{i,0}^{t-\tau_i} - y_{i,q}^{t-\tau_i} \right\|^2$$

$$\leq \mathbb{E} \left\| \sum_{l=0}^{q-1} \eta_l^{(l)} g_i \left( y_{i,l}^{t-\tau_i} \right) \right\|^2 \leqslant 3q \left( \sum_{l=0}^{q-1} \left( \eta_l^{(l)} \right)^2 \right) \left( \sigma_l^2 + \sigma_g^2 + G \right). \tag{21}$$

Thus, the upper bound on $T_3$ becomes:

$$
\begin{aligned}
\mathbb{E}\left[T_3\right] &\leqslant 6\left(L^2\eta_g^2 Q\tau_{max,k}^2\left(\sum_{i=0}^{Q-1}\left(\eta_l^{(l)}\right)^2\right)\left(\sigma_l^2+\sigma_g^2+G\right)+L^2 q\left(\sum_{i=0}^{q-1}\left(\eta_l^{(l)}\right)^2\right)\left(\sigma_l^2+\sigma_g^2+G\right)\right)\\
&\leqslant 6L^2\left(\sum_{i=0}^{Q-1}\left(\eta_l^{(l)}\right)^2\right)\left(\eta_g^2 Q\tau_{max,k}^2+q\right)\left(\sigma_l^2+\sigma_g^2+G\right)\\
&\leqslant 6L^2 Q\left(\sum_{i=0}^{Q-1}\left(\eta_l^{(l)}\right)^2\right)\left(\eta_g^2\tau_{max,k}^2+1\right)\left(\sigma_l^2+\sigma_g^2+G\right)
\end{aligned}
\tag{22}
$$

Inserting the upper bound on $T_3$ into Eqn.( 16), we have,

$$
\mathbb{E}\left[T_1\right] \leq \frac{-\eta_g}{2}\left(\sum_{q=0}^{Q-1}\eta_1^{(q)}\right)\left\|\nabla f\left(w^t\right)\right\|^2+\sum_{q=0}^{Q-1}\frac{\eta_g\eta_l^{(q)}}{2}\mathbb{E}\left[T_3\right]-\sum_{q=0}^{Q-1}\frac{\eta_g\eta_l^{(q)}}{2}\mathbb{E}_{\mathcal{H}}\|\frac{1}{m}\sum_{i=1}^m\nabla F_i\left(y_{i,q}^{t-\tau_i}\right)\|^2
\tag{23}
$$

Let $\alpha_1(Q):=\sum_{q=0}^{Q-1}\eta_l^{(q)}$ and $\alpha_2(Q):=\sum_{q=0}^{Q-1}(\eta_l^{(q)})^2$. Then

$$
\begin{aligned}
\mathbb{E}\left[T_1\right] \leq{} &\frac{-\eta_g\alpha_1(Q)}{2}\|\nabla f(w^t)\|^2+3\eta_g L^2 Q\alpha_1(Q)\alpha_2(Q)\left(\eta^2\tau_{max,K}^2+1\right)\left(\sigma_l^2+\sigma_g^2+G\right)\\
&\underbrace{-\sum_{q=0}^{Q-1}\frac{\eta_g\eta_l^{(q)}}{2}\mathbb{E}_{\mathcal{H}}\left\|\frac{1}{m}\sum_{i=1}^m\nabla F_i(y_{i,q}^{t-\tau})\right\|^2}_{T_4}
\end{aligned}
\tag{24}
$$

To derive the upperbound on the R.H.S. of Eqn.( 13), we now need to upper bound $\mathbb{E}[T_2]$. We proceed by adding and subtracting the expected gradient within the norm,

$$
\begin{aligned}
\mathbb{E}\left[T_2\right] &= \mathbb{E}\left[\frac{L\eta_g^2}{2k^2}\left\|\sum_{k\in S_t}\sum_{q=0}^{Q-1}\eta_l^{(q)}g_k\left(y_{k,q}^{t-\tau_k}\right)\right\|^2\right]\\
&\leq \frac{L\eta_g^2\alpha_2(Q)\sigma_l^2}{2}+\underbrace{\frac{LQ\eta_g^2}{2m}\sum_{q=0}^{Q-1}\sum_{i=1}^m(\eta_l^{(q)})^2\mathbb{E}_{\mathcal{H}}\left[\|\nabla F_i\left(y_{i,q}^{t-\tau_i}\|^2\right]\right.}_{T_5}
\end{aligned}
\tag{25}
$$

We now show that $T_4+T_5\leq 0$

$$
\begin{aligned}
(T_4+T_5) &= -\sum_{q=0}^{Q-1}\frac{\eta_g\eta_l^{(q)}}{2}\mathbb{E}_{\mathcal{H}}\left\|\frac{1}{m}\sum_{i=1}^m\nabla F_i(y_{i,q}^{t-\tau})\right\|^2+\frac{LQ\eta_g^2}{2m}\sum_{q=0}^{Q-1}\sum_{i=1}^m(\eta_l^{(q)})^2\mathbb{E}_{\mathcal{H}}\left[\|\nabla F_i\left(y_{i,q}^{t-\tau_i}\|^2\right]\right.\\
&= \sum_{q=0}^{Q-1}\sum_{i=1}^m\left(-\frac{\eta_g\eta_l^{(q)}}{2m}+\frac{LQ\eta_g^2(\eta_l^{(q)})^2}{2m}\right)\mathbb{E}_{\mathcal{H}}\left\|\nabla F_i(y_{i,q}^{t-\tau})\right\|^2
\end{aligned}
\tag{26}
$$

To ensure $T_4+T_5\leq 0$, it is sufficient to choose $\eta_g\eta_l^{(q)}\leq\frac{1}{L}$ for all local steps $q=0,...,Q-1$. Now, plugging (24), (25), and (26) into (13),

$$
\begin{aligned}
\mathbb{E}[f\left(w^{t+1}\right)] \leq{} &\mathbb{E}\left[f\left(w^t\right)\right]-\frac{\eta_g\alpha_1(Q)}{2}\left\|\nabla f\left(w^t\right)\right\|^2+3\eta_g L^2 Q\alpha_1(Q)\alpha_2(Q)\left(\eta_g^2\tau_{max,K}+1\right)\left(\sigma_1^2+\sigma_g^2+G\right)\\
&+\frac{L}{2}\eta_g^2\alpha_2(Q)\sigma_l^2
\end{aligned}
\tag{27}
$$

Summing up $t$ from $1$ to $T$ and rearrange, yields

$$\sum_{t=0}^{T-1} \eta_g \alpha_1(Q) \left\| \nabla_f \left( w^t \right) \right\|^2 \leq \sum_{t=0}^{T-1} 2 \left( \mathbb{E} \left[ f \left( w^t \right) \right] - \mathbb{E} \left[ f \left( w^{t+1} \right) \right] \right)$$

$$+ 3 \sum_{t=0}^{T-1} \eta_g L^2 Q \alpha_1(Q) \alpha_2(Q) \left( \eta_g^2 \tau_{max,K}^2 + 1 \right) \left( \sigma_1^2 + \sigma_g^2 + G \right)$$

$$+ \frac{L}{2} \eta_g^2 \alpha_2(Q) \sigma_l^2$$

$$\leq 2 \left( f \left( w^0 \right) - f \left( w^* \right) \right) + 3 \sum_{t=0}^{T-1} \eta_g L^2 \alpha_1(Q) \alpha_2(Q) \left( \eta_g^2 \tau_{max,k}^2 + Q \right) \left( \sigma_l^2 + \sigma_g^2 + G \right)$$

$$+ \frac{L}{2} \eta_g^2 \alpha_2(Q) \sigma_l^2 \tag{28}$$

Thus we have

$$\frac{1}{T} \sum_{t=0}^{T-1} \left\| \nabla f \left( w^t \right) \right\|^2 \leqslant \frac{2 \left( f \left( w^0 \right) - f \left( w^* \right) \right)}{\eta_g \alpha_1(Q) T}$$

$$+ 3 L^2 Q \alpha_2(Q) \left( \eta_g^2 \tau_{max,k} + 1 \right) \left( \sigma_l^2 + \sigma_g^2 + G \right) + \frac{L}{2} \frac{\eta_g \alpha_2(Q)}{\alpha_1(Q)} \sigma_l^2 \tag{29}$$

For sufficiently large $T$, the algorithm achieves the convergence rate as shown in Equation 29. As we can see, the convergence guarantee increases with increasing $K$ as we tend to go closer to the synchronous aggregation. Also, as non-IID bias increases, gradient variances increase, and weakens the convergence guarantee.

