# OpenReview forum: "Kuiper: Moderated Asynchronous Federated Learning on Heterogeneous Mobile Devices with Non-IID Data"
_ICLR.cc/2023/Conference — Submitted to ICLR 2023_

### Official Review · Reviewer_t9vz · 2022-10-26

**Confidence:** 3
**Correctness:** 2
**Technical Novelty And Significance:** 1
**Empirical Novelty And Significance:** 1
**Recommendation:** 3

**Clarity, Quality, Novelty And Reproducibility:**

There are some rooms for improving the quality, clarity and originality of the proposed method, as mentioned above.


**Strength And Weaknesses:**

**Strength**
- They tackle the practical federated learning where asynchronous communication is frequent due to the computational heterogeneity.
- The results are impressive.

**Weaknesses**
- The critical limitation of this work is the lack of significance. The proposed method for calculating staleness seems a rule-based algorithm which is hard to consider as significant. I think it could be a better way to devise a neural model learning the staleness and importance of asynchronously updated local models (neural aggregator, etc).
- Error bars are missing and thus it is hard to discuss the marginal case i.e. Fig 4 (b)
- Minor:
    - Paper organization need to be polished (inbetween spaces are too tight)


**Summary Of The Paper:**

The authors tackle a practical problem of federated learning where asynchronous communication is frequent due to the computational heterogeneity. The authors propose a method considering taleness of local updates when performing model aggregation. They demonstrate the effectiveness of their method.


**Summary Of The Review:**

I enjoyed reading the paper, but several improvements are required.

---

> ### Author Response · Authors · 2022-11-12
> **Novelty**
>
> 1. We respectfully disagree with the reviewer. Our contribution is that we provide a solution to do federated learning with heterogeneous devices, including embedded devices. Our algorithm for the second time balances device capability and value of the data each device has. We quantitatively show our superiority to the first work that attempted this and which appeared at the top systems conference, OSDI 2021. We then show that our approach enables for the first time a heavyweight task, human activity recognition, on embedded devices. \
> The calculation of staleness is a small detail in our algorithm and is not a contribution (nor do we claim that in the submission). We would posit that a neural network to learn the staleness is an overkill and would be vulnerable to the weakness of overfitting.
>
> 2. In Fig. 4b, as the number of clients increases (say to 50 and beyond), the number of training samples per client per class decreases to a point where it gets really difficult to train a any model with such a low number of samples. In the marginal cases where the number of clients is high (50, 100), all the methods achieve the “random guess” accuracy, that is, the accuracy we get when we randomly guess ($\frac{1}{number\_of\_classes}$). Also, the error bar would have helped, but not in this case.\
> We are attaching plots with error bars for your reference.\
> Fig. 4a: https://ibb.co/CmX2qrj \
> Fig. 4b: https://ibb.co/qkZnJFC

---

> > ### Author Response · Authors · 2022-11-18
> > **Contributions and Novelty**
> >
> > We would like to thank you for reviewing our paper. We have strived to address your queries. Since the discussion period is quickly drawing to a close, we would love to address any other queries or clarifications you may have.
> >
> > Our novelty lies in rewarding and penalizing the clients depending on the usefulness of their data to the model training and the staleness of the update. Oort calculates the utility score depending on how stale they are and their loss value. After selecting the clients using the utility score, they are aggregated using FedAVG. Oort does not scale the individual gradients; thus, the selected clients contribute equally to the global model. This turns out to be a critical flaw with heterogeneous clients, and especially so under non-IID settings.\
> > FedBuff considers the staleness parameter and penalizes the client accordingly but does not consider the gradients' quality or usefulness. It does not have a client selection policy or a gradient weighting policy according to the clients' performance; thus, the model is biased towards the fast clients' data distribution. When the non-IID bias is high, some slow clients may have exclusive data, which is important for the overall training and thus it is needed to aggregate that client’s model with high importance. Our algorithm does that, whereas FedBuff does not.\
> > While in hindsight, the two advances over Oort and FedBuff (one each) may appear intuitive, they are fundamental. We bring out both theoretically and empirically the profound impact of these two innovations. Another dimension of novelty is that we bring together the advantages of asynchronous FL (being able to keep training efficient with heterogeneous clients) with those of synchronous FL (being able to cluster clients by their quality, considering updates that are close in time). \
> > The improvement mentioned above is very well shown in the 4a (IID setup) and 4b (non-IID Setup).

---

> > > ### Author Response · Authors · 2022-12-02
> > > **More queries**
> > >
> > > We would like to thank you for reviewing our paper. We have strived to address your queries. Since the discussion period is quickly drawing to a close, we would love to address any other queries or clarifications you may have.

---

### Official Review · Reviewer_cvzu · 2022-10-28

**Confidence:** 3
**Correctness:** 3
**Technical Novelty And Significance:** 2
**Empirical Novelty And Significance:** 3
**Recommendation:** 6

**Clarity, Quality, Novelty And Reproducibility:**

Thanks for the submission -- this work appears to make contributions in exploring how best to incorporate non-IID updates in an asynchronous fashion.   There were some areas where I had additional questions / confusion.

Figure 1 is greatly appreciated, but I was unable to rectify that figure and the description of the update policy.   In particular, Section 3 don't appear to define what t or \tau (or \tau_{i}) are.  In particular, are they "indices" of the t-th round?  Or are they actual measurements of latency?   If you look at the figure, each arrow representing t-tau appears to be the time between updates for each client.  That means t and tau are in terms of time.    In the figure, is t from t_2 or t_3?  Then we look at eq 2.  T_0 is a threshold, so assume it's a point in time.  But here t is the t-th iteration?  Why is t the same across clients?   Equation 3 raises similar issues.  s() makes sense if s and tau^{ct} are update index, but not if they are clock time (t continues to grow while the average delay does not).

The paper does a nice job making sure that relevant work is cited.  However, it was difficult to figure out exactly how Kuiper is different from its most closely related systems (Oort/FedBuff).   One confusion arises where Oort is described in the Async FL systems (Sec 2 P2).   But then is described as synchronous in sec 5 burst size paragraph.   There is language describing how Oort also measures client utility, but never it is precise enough to compare to Kuiper's approach.   Sec 2 P2 says Oort "waits for a fixed K to 'synchronize'" -- sounds just like Kuiper.  Then sec 5 baseline comparison says that Oort "gives the same weight to all client updates" -- not sure what that means exactly.
Then in the burst size paragraph it says Oort "waits for the K chosen clients in each epoch."   The paper should precisely describe the competing system in one place to make it utterly transparent how Kuiper is different.

Sometimes the work seems confused about its ultimate contribution.  Is it for doing action recognition at edge devices or is it about handling stragglers when doing FL?  This happens in Section 5 where the first question that comes to mind is whether that task can run on edge nodes.  I had no idea that was the *main* point of the paper.  The last question is whether it handles stragglers.  It's also the very smallest graphs in the paper.  You should refer to appendix for the alpha/beta question.


Lesser items:

* Including KD in design was confusing.  The paper reads as if KD creates the initial scaled-down edge models, and then Kuiper (and competing systems) are used to "fine tune" the client models.   In this case, what does KD have to do at all with the system design?  It seems like it would only really impact your implementation and experiments.  One doesn't need KD to use Kuiper, right?

* Sec 2 P2 and P3 also have nearly the same content.   P2 ends with issues with non-IID then P3 starts by saying the same thing.

* Sec1 page 2, footnote 3, you mean "Kuper's superiority" not "our superiority" right? :)

* The paper treats the prediction task as more than just a good, complex model to train.   The "Takeaways" section says this is the first time we can run action recognition on edge devices, but we could have just installed the centrally trained model.  In addition, while I'd agree that the 9-10% gains are meaningful, it's not because the current central accuracy is 47%.  47% is still awful - FL techniques won't improve upon centrally trained model performance.

* The paper describes averaging and merging client gradients but eqs 2,3,4 all use weights w.  I don't think it invalidates the work, but it's not consistent.

* Sec 2 P3 says highly-skewed distributions happen in mobile computing environments.   What's the basis for that statement?  Are we talking about mobile computing or edge computing here (experiments seemed to be edge devices)?


**Strength And Weaknesses:**

Positives
+ Provides a delta on top of existing work in sync/async buffered approaches for dealing with stragglers.
+ Extensive experiments on a challenging modeling task across multiple datasets against competing systems
+ Experiments showed training/time benefits, and explored system design (scale, tuning variables)

Negatives
- The description of the algorithm, particularly the use/measure of time, is not clear
- While the system compares to Oort / FedBuff, the differences aren't clear and spread across the paper
- The writing quality and organization makes understanding the contributions and system challenging.

**Summary Of The Paper:**

Kuiper is a federated learning system designed to produce accurate global models in the presence of non-IID client data.   In particular Kuiper addresses the straggling clients with an asynchronous, buffered aggregation technique.   Kuiper proceeds in rounds, starting a round when K client updates arrive.  It then adjusts the global model based on client data set size, update latency, and local training error.  The work provides a convergence analysis and experiments show that the system provides comparable or better prediction performance against recent work on different video action recognition benchmarks.

**Summary Of The Review:**

Moderate contribution in asynchronous FL.   Strong evaluation / extensive results on a non-trivial problem across datasets, against competing algorithms, including exploring various tuning variables.   The writing and paper focus makes understanding details of the technical contribution difficult, as well as placing it in the context of the systems to which they compare.

[Post Author Response].  I'd like to thank the author's for explaining the questions my review raised.  The work unifyies techniques from other systems in a non-trivial manner, which is an important contribution for the community.   The authors address many of the presentation problems, and the system performance shows clear global model and training time improvements.    In view of this I'm adjusting the empirical significance and recommendation to a 6.

---

> ### Author Response · Authors · 2022-11-12
> **Algorithm Clarity, Novelty**
>
> Thank you for your valuable feedback.
>
> ## Clarity
> I am sorry for the confusion.
>
> 1. $\tau_i$ is the global iteration when $i^{th}$ client started the training. For example, client 'i' started computation in the $k1^{th}$ global iteration, then $\tau_i$ = k1, and finished in the k2th global iteration, then t- $\tau_i$ for that client will be k2-k1. T-$\tau_i$ is the staleness factor. In this example,  client 1 and 4 have completed their global update in one iteration from t2 to t3, whereas client 3 has taken 2 global iterations (from t1 to t3). Thus the update from client 3 is staled by 1 extra iteration. So t-$\tau_i$ is 1 for client 1 and 4 and 2 for client 3. \
> $T_{0}$ is in terms of global iteration and not absolute time. We have set it to 10.
> Here, time, t, and $\tau_i$, $\tau_{ct}$ are in terms of global iterations. \
> We will make it more clear in the final draft.
>
> 2. Our novelty lies in rewarding and penalizing the clients depending on the usefulness of their data to the model training and the staleness of the update. Oort calculates the utility score depending on how stale they are and their loss value. After selecting the clients using the utility score, they are aggregated using FedAVG. Oort does not scale the individual gradients; thus, the selected clients contribute equally to the global model. This turns out to be a critical flaw with heterogeneous clients, and especially so under non-IID settings.
> FedBuff considers the staleness parameter and penalizes the client accordingly but does not consider the gradients' quality or usefulness. It does not have a client selection policy or a gradient weighting policy according to the clients' performance; thus, the model is biased towards the fast clients' data distribution. When the non-IID bias is high, some slow clients may have exclusive data, which is important for the overall training and thus it is needed to aggregate that client’s model with high importance. Our algorithm does that, whereas FedBuff does not.
> While in hindsight, the two advances over Oort and FedBuff (one each) may appear intuitive, they are fundamental. We bring out both theoretically and empirically the profound impact of these two innovations. Another dimension of novelty is that we bring together the advantages of asynchronous FL (being able to keep training efficient with heterogeneous clients) with those of synchronous FL (being able to cluster clients by their quality, considering updates that are close in time).
> Our algorithm is generic and is designed to handle stragglers in a high non-IID data-distribution setting. We have demonstrated the effectiveness of our algorithm by creating a real-world heterogeneous setup by utilizing different Nvidia devices with varying computing power. This heterogeneity is further stretched if we use a heavy task like video action recognition. Nonetheless, we show that our algorithm is scalable across several devices (10,25,50,100,1000) and also for different tasks like classification (CIFAR10, MNIST, FMNIST), Video action recognition (UCF101, HMDB51), Next character prediction (Shakespeare) with varying non-IID degree from IID (0) to completely non-IID (1).
>
> 3. Our algorithm is generic and is designed to handle stragglers in a high non-IID data-distribution setting. We have demonstrated the effectiveness of our algorithm by creating a real-world heterogeneous setup by utilizing different Nvidia devices with varying computing power. This heterogeneity is further stretched if we use a heavy task like video action recognition. Nonetheless, we show that our algorithm is scalable across several devices (10,25,50,100,1000) and also for different tasks like classification (CIFAR10, MNIST, FMNIST), Video action recognition (UCF101, HMDB51), Next character prediction (Shakespeare) with varying non-IID degree from IID (0) to completely non-IID (1).

---

> > ### Author Response · Authors · 2022-11-12
> > **Minor points**
> >
> >
> > 1. Yes, one can use Kuiper without Knowledge Distillation. With a high degree of non-IID, it is tough to train a model with fewer samples per class per client. Thus Knowledge Distillation helps to initialize the model from a good point, helping it to converge better.
> > 2. We try to put the problem with asynchronous setup in paragraph 2 and how non-IID data amplifies them. Then listing the contribution of previous methods to solve that issue. In paragraph 3, we describe our method and why we are better than the previous works.  We agree that the content is overlapping, and we will take care of it in the final version.
> > 3. Because of memory issues on edge devices, we used a frame rate of 8 per video. That’s why the accuracy is so low.
> > 4. As we have the initial weights and final weights and we know the learning rate, we can calculate the gradients. So working with either weights or gradients is fine. But we will make it consistent in the final version.
> > 5. We have used edge devices in our experiments. Here, we want to say that data collection is different in different scenarios, and thus the gathered data varies a lot among the clients.

---

> > > ### Author Response · Authors · 2022-11-18
> > > **Other queries**
> > >
> > > We would like to thank you for reviewing our paper. We have strived to address your queries. Since the discussion period is quickly drawing to a close, we would love to address any other queries or clarifications you may have.

---

### Official Review · Reviewer_x4EL · 2022-11-03

**Confidence:** 4
**Clarity, Quality, Novelty And Reproducibility:** 1. I would suggest the authors to rew…
**Correctness:** 4
**Technical Novelty And Significance:** 2
**Empirical Novelty And Significance:** 3
**Recommendation:** 5

**Strength And Weaknesses:**

## Strength
1. The paper is easy to read and follow. The problem and motivation is clearly stated in the introduction.
2. The aggregation scheme is simple and can be easily incorporated without making any new assumptions.
3. The takeaways section helps capture the high level points of the paper.

## Weakness
1. Equation 2 is hard to follow. For example, what is $w_{new,t}^i$ ?
2. The video action recognition experiments were done on a very small number of devices (less than 12). FedBuff is design for cross-device setting with very high concurrency and K to be a small fraction of concurrency.
3. In large scale experiments (Figure 5), the difference between Kuiper vs Oort and FedBuff seems negligible. Did the author repeat the runs for multiple trials and report the average?

**Summary Of The Paper:**

## The problem
The paper introduces an asynchronous  aggregation scheme to address the straggler problem and also taking in the effect of non-iid. The paper focuses on "heavyweight" learning task such as video action recognition that are out of reach for mobile devices processing power. Overall, the paper aims to address the problem of how to best aggregate the updates sent by all clients in order to maximize information
learned while minimizing any adverse effect from slow updates.
## The solution
Scale client update based on its staleness and the client's local training loss. The aggregation scheme gives more importance to the clients with low training accuracy to push to global model to learn from clients with low accuracy.




**Summary Of The Review:**

Overall, I believe this paper is a simple extension to FedBuff. However, I think the improvement is incremental and the problem of FL on heavyweight ML task seems like a niche problem space.

## Post-rebuttal update
The authors answered my concerns about scalability and clarified the difference between KUIPER and FedBuff. However, I think the paper can benefit from throughout revision to make it clear what the contributions and focuses of the paper are. Secondly, the paper lacks details about the how the client execution time, staleness and how stragglers are simulated. I appreciate that the authors did clarify these point in their comments.

I will stand by my initial rating of below the acceptance threshold. I think this is good contribution to the community once the weaknesses are addressed.

---

> ### Author Response · Authors · 2022-11-12
> **Scalability, Device Setup**
>
> 1. $w^i_{new, t}$ is the new update from $i^{th}$ client in $t^{th}$ iteration.
> 2. HMDB51 and UCF101 are small datasets. With a high number of clients (100 clients), each client gets a small number of samples per class (1-2 samples) and is thus not trainable better than random guess (as mentioned in the Scalability part of the Experimental Evaluation section, Fig 4b). But we have still shown the experiments on the Video Action Recognition task up to 100 devices. We have demonstrated our scalability by experimenting on 1000 devices on other tasks (Classification, Next character prediction).
> 3. We have good improvement (11% faster convergence compared to Oort [OSDI-21], up to 12% and 9% improvement in test accuracy compared to FedBuff [AISTAT-22] and Oort [OSDI-21] on HMDB51, and 10% and 9% on UCF101) for video action recognition, a comparatively heavy task. On other datasets, we are showing the scalability of our algorithm on some light tasks like classification on MNIST-CIFAR10. We still have the edge over the other methods. We repeated the experiments for 3 trials.
>
> ## Clarity
> 1. Yes, that is correct. We will update it in the final version.
> 2. FedBuff focuses on waiting to form a cluster and then aggregating those cluster clients together. FedBuff weighs clients in the cluster according to their delay, i.e., how fresh or stale their updates are. It does not consider the quality of data the client has.
> We are doing things differently:\
> a. Weighing the complete cluster according to the staleness\
> b. Weighing different clients in the cluster differently according to their training progress.\
> Also, FedBuff gives greater weight to the fast clients (because of more frequent updates). It does not have a client selection policy or a gradient weighing policy according to their performance. Thus, the model is biased toward the fast clients' data distribution. When the non-IID bias is high, some slow clients will have exclusive data, which is important for the overall training and thus need to aggregate that client’s model with high importance. Our algorithm does that, whereas FedBuff does not.
> 3. In some cases, edge devices (IoT) and video capturing devices are used to process the video and smartly store the useful part or quick inference. Studying the heavyweight ML tasks, specifically FL on edge is useful.\
> [1] D. Pudasaini and A. Abhari, "Scalable Object Detection, Tracking and Pattern Recognition Model Using Edge Computing," 2020 Spring Simulation Conference (SpringSim), 2020, pp. 1-11, doi: 10.22360/SpringSim.2020.CNS.003.\
> [2] Wan, Shaohua, Songtao Ding, and Chen Chen. "Edge computing enabled video segmentation for real-time traffic monitoring in internet of vehicles." Pattern Recognition 121 (2022): 108146.
> 4. Knowledge distillation enables smaller networks to learn more complex things from the teacher counterparts they wouldn’t have learnt on their own. This is more effective than simple pre-training. The convergence becomes trickier in a Federated Learning setting with a higher non-IID degree. A good initialization like this helps us achieve a good convergence.

---

### Decision · Program_Chairs · 2023-01-20

**Decision:**

Reject

**Justification For Why Not Higher Score:**

Given the current reviews and discussion, my recommendation is to reject this paper. It wasn't discussed, but my sense is that the discussion wouldn't have changed anything.

There is a possible argument for raising this to Accept (poster). That would be to (a) discount the input of Review t9vz, which is both the lowest score and also somewhat lower confidence, and (b) to put a lot of faith in the authors to accurately implement all of the changes and clarifications that came about based on feedback from the reviewers.

However, the number of changes required to improve the presentation to be acceptable is substantial enough that they should be reviewed again before the paper is accepted. If the authors had uploaded a revision during the rebuttal period, then the reviewers and AC could have seen the changes and made a more confident call. Since no revision was uploaded (as far as I can tell, the most recent version was uploaded in October), this isn't an option.

**Justification For Why Not Lower Score:**

N/A

**Metareview: Summary, Strengths And Weaknesses:**

This paper proposes an approach to asynchronous federated learning. The focus is on FL or video action recognition, and the paper proposes a new approach aggregating client responses in async FL. Experiment results suggest the proposed approach outperforms previous approaches FedAsync, FedBuff, and Oort.

One major weakness of the initial submission was the clarity of presentation in the paper. All reviewers raised concerns about this and difficulty of understanding the difference between the proposed approach and previous work, as well as understanding what constitutes the main contribution of this paper. The author responses helped to address some of these concerns, especially for one reviewer who raised their score. However the issues remained for other reviewers. Although the responses helped with clarifications, it doesn't appear that any revised version of the paper was updated during the rebuttal period, and it would have been very helpful to see concretely how the paper would be changed to address these concerns.

Other concerns were raised about missing details of the experimental setup. These were somewhat clarified in the rebuttal, but again it would have helped to see a revised manuscript implementing all of the changes.

**Summary Of Ac-Reviewer Meeting:**

n/a